# Preconditioned Gradient Descent for Over-Parameterized Nonconvex Matrix Factorization

**Gavin Zhang**
University of Illinois at Urbana–Champaign
jialun2@illinois.edu

**Salar Fattahi**
University of Michigan
fattahi@umich.edu

**Richard Y. Zhang**
University of Illinois at Urbana–Champaign
ryz@illinois.edu

## Abstract

In practical instances of nonconvex matrix factorization, the rank of the true solution $r^\star$ is often unknown, so the rank $r$ of the model can be overspecified as $r > r^\star$. This over-parameterized regime of matrix factorization significantly slows down the convergence of local search algorithms, from a linear rate with $r = r^\star$ to a sublinear rate when $r > r^\star$. We propose an inexpensive preconditioner for the matrix sensing variant of nonconvex matrix factorization that restores the convergence rate of gradient descent back to linear, even in the over-parameterized case, while also making it agnostic to possible ill-conditioning in the ground truth. Classical gradient descent in a neighborhood of the solution slows down due to the need for the model matrix factor to become singular. Our key result is that this singularity can be corrected by $\ell_2$ regularization with a specific range of values for the damping parameter. In fact, a good damping parameter can be inexpensively estimated from the current iterate. The resulting algorithm, which we call preconditioned gradient descent or PrecGD, is stable under noise, and converges linearly to an information theoretically optimal error bound. Our numerical experiments find that PrecGD works equally well in restoring the linear convergence of other variants of nonconvex matrix factorization in the over-parameterized regime.

## 1   Introduction

Numerous problems in machine learning can be reduced to the *matrix factorization* problem of recovering a low-rank positive semidefinite matrix $M^\star \succeq 0$, given a small number of potentially noisy observations [1–7]. In every case, the most common approach is to formulate an $n \times n$ candidate matrix $M = XX^T$ in factored form, and to minimize a *nonconvex* empirical loss $f(X)$ over its $n \times r$ low-rank factor $X$. But in most real applications of nonconvex matrix factorization, the rank of the ground truth $r^\star = \mathrm{rank}(M^\star)$ is unknown. It is reasonable to choose the rank $r$ of the model $XX^T$ conservatively, setting it to be potentially larger than $r^\star$, given that the ground truth can be exactly recovered so long as $r \geq r^\star$. In practice, this will often lead to an *over-parameterized* regime, in which $r > r^\star$, and we have specified more degrees of freedom in our model $XX^T$ than exists in the underlying ground truth $M^\star$.

Zhuo et al. [8] recently pointed out that nonconvex matrix factorization becomes substantially less efficient in the over-parameterized regime. For the prototypical instance of matrix factorization known as *matrix sensing* (see Section 3 below for details) it is well-known that, if $r = r^\star$, then (classic) gradient descent or GD

$$X_{k+1} = X_k - \alpha \nabla f(X_k) \tag{GD}$$

converges at a linear rate, to an $\epsilon$-accurate iterate in $O(\kappa \log(1/\epsilon))$ iterations, where $\kappa = \lambda_1(M^\star)/\lambda_{r^*}(M^\star)$ is the condition number of the ground truth [9, 10]. But in the case that $r > r^\star$, Zhuo et al. [8] proved that gradient descent slows down to a *sublinear* convergence rate, now requiring $\text{poly}(1/\epsilon)$ iterations to yield a comparable $\epsilon$-accurate solution. This is a dramatic, exponential slow-down: whereas 10 digits of accuracy can be expected in a just few hundred iterations when $r = r^\star$, tens of thousands of iterations might produce just 1-2 accurate digits once $r > r^\star$. The slow-down occurs even if $r$ is just off by one, as in $r = r^\star + 1$.

It is helpful to understand this pheonomenon by viewing over-parameterization as a special, extreme case of ill-conditioning, where the condition number of the ground truth, $\kappa$, is taken to infinity. In this limit, the classic linear rate $O(\kappa \log(1/\epsilon))$ breaks down, and in reality, the convergence rate deteriorates to sublinear.

In this paper, we present an inexpensive *preconditioner* for gradient descent. The resulting algorithm, which we call PrecGD, corrects for both ill-conditioning and over-parameterization at the same time, without viewing them as distinct concepts. We prove, for the matrix sensing variant of nonconvex matrix factorization, that the preconditioner restores the convergence rate of gradient descent back to linear, even in the over-parameterized case, while also making it agnostic to possible ill-conditioning in the ground truth. Moreover, PrecGD maintains a similar per-iteration cost to regular gradient descent, is stable under noise, and converges linearly to an information theoretically optimal error bound.

We also perform numerical experiments on other variants of nonconvex matrix factorization, with different choices of the empirical loss function $f$. In particular, we consider different $\ell_p$ norms with $1 \leq p < 2$, in order to gauge the effectiveness of PrecGD for increasingly nonsmooth loss functions. Our numerical experiments find that, if regular gradient descent is capable of converging quickly when the rank is known $r = r^\star$, then PrecGD restores this rapid converging behavior when $r > r^\star$. PrecGD is able to overcome ill-conditioning in the ground truth, and converge reliably without exhibiting sporadic behavior.

## 2 Proposed Algorithm: Preconditioned Gradient Descent

Our preconditioner is inspired by a recent work of Tong et al. [11] on matrix sensing with an ill-conditioned ground truth $M^\star$. Over-parameterization can be viewed as the limit of this regime, in which $\lambda_r(M^\star)$, the $r$-th largest eigenvalue of $M^\star$, is allowed to approach all the way to zero. For finite but potentially very small values of $\lambda_r(M^\star) > 0$, Tong et al. [11] suggests the following iterations, which they named *scaled* gradient descent or ScaledGD:

$$X_{k+1} = X_k - \alpha \nabla f(X_k)(X_k^T X_k)^{-1}. \qquad \text{(ScaledGD)}$$

They prove that the scaling allows the iteration to make a large, constant amount of progress at every iteration, independent of the value of $\lambda_r(M^\star) > 0$. However, applying this same scheme to the over-parameterized case with $\lambda_r(M^\star) = 0$ results in an inconsistent, sporadic behavior.

The issues encountered by both regular GD and ScaledGD with over-parameterization $r > r^\star$ can be explained by the fact that our iterate $X_k$ must necessarily become *singular* as our rank-$r$ model $X_k X_k^T$ converges towards the rank-$r^\star$ ground truth $M^\star$. For GD, this singularity causes the per-iteration progress itself to decay, so that more and more iterations are required for each fixed amount of progress. ScaledGD corrects for this decay in per-iteration progress by suitably rescaling the search direction. However, the rescaling itself requires inverting a near-singular matrix, which causes algorithm to take on sporadic values.

A classical remedy to issues posed by singular matrices is $\ell_2$ regularization, in which the singular matrix is made "less singular" by adding a small identity perturbation. Applying this idea to ScaledGD yields the following iterations

$$X_{k+1} = X_k - \alpha \nabla f(X_k)(X_k^T X_k + \eta_k I_r)^{-1}, \qquad \text{(PrecGD)}$$

where $\eta_k \geq 0$ is the *damping* parameter specific to the $k$-th iteration. There are several interpretations to this scheme, but the most helpful is to view $\eta$ as a parameter that allows us to interpolate between ScaledGD (with $\eta = 0$) and regular GD (in the limit $\eta \to \infty$). In this paper, we prove for matrix sensing that, if the $k$-th damping parameter $\eta_k$ is chosen within a constant factor of the error

$$C_{\text{lb}} \|X_k X_k^T - M^\star\|_F \leq \eta_k \leq C_{\text{ub}} \|X_k X_k^T - M^\star\|_F, \quad \text{where } C_{\text{lb}}, C_{\text{ub}} > 0 \text{ are abs. const.} \quad (1)$$

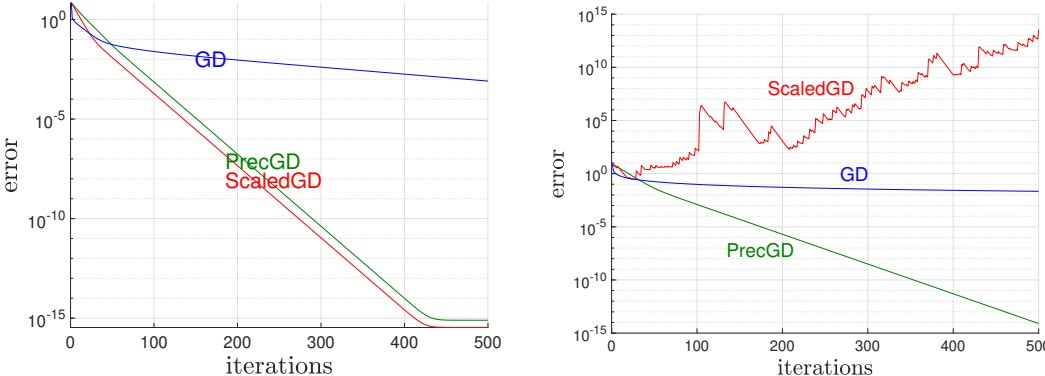

Figure 1: **PrecGD converges linearly in the overparameterized regime.** Convergence of regular gradient descent (GD), ScaledGD and PrecGD for noiseless matrix sensing (with data taken from [12, 13]) from the same initial points and using the same learning rate $\alpha = 2 \times 10^{-2}$. **(Left $r = r^*$)** Set $n = 4$ and $r^* = r = 2$. All three methods convergence at a linear rate, though GD converges at a slower rate due to ill-conditioning in the ground truth. **(Right $r > r^*$)** With $n = 4$, $r = 4$ and $r^* = 2$, over-parameterization causes gradient descent to slow down to a sublinear rate. ScaledGD also behaves sporadically. Only PrecGD converges linearly to the ground truth.

then the resulting iterations are guaranteed to converge linearly, at a rate that is independent of both over-parameterization and ill-conditioning in the ground truth $M^*$. With noisy measurements, setting $\eta_k$ to satisfy (1) will allow the iterations to converge to an error bound that is well-known to be minimax optimal up to logarithmic factors [14].

We refer to the resulting iterations (with a properly chosen $\eta_k$) as *preconditioned* gradient descent, or PrecGD for short. For matrix sensing with noiseless measurements, an optimal $\eta_k$ that satisfies the condition (1) is obtained for free by setting $\eta_k = \sqrt{f(X_k)}$. In the case of noisy measurements, we show that a good choice of $\eta_k$ is available based on an approximation of the noise variance.

## 3 Background and Related Work

**Notations.** We use $\|\cdot\|_F$ to denote the Frobenius norm of a matrix and $\langle\cdot,\cdot\rangle$ is the corresponding inner product. We use $\gtrsim$ to denote an inequality that hides a constant factor. The big-O notation $\tilde{O}$ hides logarithimic factors. The gradient of the objective is denoted by $\nabla f(X) \in \mathbb{R}^{n\times r}$. The eigenvalues are assumed to be in decreasing order: $\lambda_1 \geq \lambda_2 \geq \cdots \geq \lambda_r$.

The symmetric, linear variant of matrix factorization known as matrix sensing aims to recover a positive semidefinite, rank-$r^*$ ground truth matrix $M^*$, from a small number $m$ of possibly noisy measurements

$$y = \mathcal{A}(M^*) + \epsilon, \qquad \text{where } \mathcal{A}(M^*) = [\langle A_1, M^*\rangle, \langle A_2, M^*\rangle, \ldots, \langle A_m, M^*\rangle]^T,$$

in which $\mathcal{A}$ is a linear measurement operator, and the length-$m$ vector $\epsilon$ models the unknown measurement noise. A distinguishing feature of matrix sensing is that $\mathcal{A}$ is assumed to satisfy the *restricted isometry property* [14, 15]. Throughout this paper, we will always assume that $\mathcal{A}$ satisfies RIP with parameters $(2r, \delta)$.

**Definition 1** (RIP). The linear operator $\mathcal{A}$ satisfies RIP with parameters $(2r, \delta)$ if there exists constants $0 \leq \delta < 1$ and $m > 0$ such that, for every rank-$2r$ matrix $M$, we have

$$(1-\delta)\|M\|_F^2 \leq \frac{1}{m}\|\mathcal{A}(M)\|^2 \leq (1+\delta)\|M\|_F^2.$$

A common approach for matrix sensing is to use a simple algorithm like gradient descent to minimize the *nonconvex* loss function:

$$f(X) = \frac{1}{m}\left\|y - \mathcal{A}(XX^T)\right\| = \frac{1}{m}\left\|\mathcal{A}(M^* - XX^T) + \epsilon\right\|^2. \tag{2}$$

Recent work has provided a theoretical explanation for the empirical success of this nonconvex approach. Two lines of work have emerged.

**Local Guarantees.** One line of work studies gradient descent initialized inside a neighborhood of the ground truth where $X_0 X_0^T \approx M^\star$ already holds [10, 16–19]. Such an initial point can be found using spectral initialization, see also [18, 20–23]. With exact rank $r = r^\star$, previous authors showed that gradient descent converges at a linear rate [9, 10]. In the over-parameterized regime, however, local restricted convexity no longer holds, so the linear convergence rate is lost. Zhuo et al. [8] showed that while spectral initialization continues to work under over-parameterization, gradient descent now slows down to a sublinear rate, but it still converges to a statistical error bound of $\tilde{\mathcal{O}}(\sigma^2 n r^\star / m)$, where $\sigma$ denotes the noise variance. This is known to be minimax optimal up to logarithmic factors [14]. In this paper, we prove that PrecGD with a damping parameter $\eta_k$ satisfying (1) also converges to an $\tilde{\mathcal{O}}(\sigma^2 n r^\star / m)$ statistical error bound.

**Global Guarantees.** A separate line of work [13, 24–31] established global properties of the landscapes of the nonconvex objective $f$ in (2) and its variants and showed that local search methods can converge globally. With exact rank $r = r^\star$, Bhojanapalli et al. [24] proved that $f$ has no spurious local minima, and that all saddles points have a strictly negative descent direction (strict saddle property [32], see also [28, 33]). In the over-parameterized regime, however, we are no longer guaranteed to recover the ground truth in polynomial time.

**Other related work.** Here we mention some other techniques can be use to solve matrix sensing in the over-parameterized regime. Classically, matrix factorization was solved via its convex SDP relaxation [14, 15, 34–36]. The resulting $\mathcal{O}(n^3)$ to $\mathcal{O}(n^6)$ time complexity [37] limits this technique to smaller problems, but these guarantees hold without prior knowledge on the true rank $r^\star$. First-order methods, such as ADMM [38–40] and soft-thresholding [41], can be used to solve these convex problems with a per-iteration complexity comparable to nonconvex gradient descent, but they likewise suffer from a sublinear convergence rate. Local recovery via spectral initialization was originally proposed for alternating minimization and other projection techniques [21, 23, 34, 42–45]. These also continue to work, though a drawback here is a higher per-iteration cost when compared to simple gradient methods. Finally, we mention a recent result of Li et al. [46], which showed in the over-parameterized regime that gradient descent with early termination enjoys an algorithmic regularization effect.

## 4  Sublinear Convergence of Gradient Descent

In order to understand how to improve gradient descent in the over-parameterized regime, we must first understand why existing methods fail. For an algorithm that moves in a search direction $D$ with step-size $\alpha$, it is a standard technique to measure the corresponding decrement in $f$ with a Taylor-like expansion

$$f(X - \alpha D) \leq f(X) - \alpha \underbrace{\langle \nabla f(X), D \rangle}_{\text{linear progress}} + \alpha^2 \underbrace{(L/2) \|D\|_F^2}_{\text{inverse step-size}} \tag{3}$$

in which $L$ is the usual gradient Lipschitz constant (see e.g. Nocedal and Wright [47, Chapter 3]). A good search direction $D$ is one that maximizes the linear progress $\langle \nabla f(X), D \rangle$ while also keeping the inverse step-size $(L/2)\|D\|_F^2$ sufficiently small in order to allow a reasonably large step to be taken. As we will show in this section, the main issue with gradient descent in the over-parameterized regime is the first term, namely, that the linear progress goes down to zero as the algorithm makes progress towards the solution.

Classical gradient descent uses the search direction $D = \nabla f(X)$. Here, a common technique is to bound the linear progress at each iteration by a condition known as *gradient dominance* (or the Polyak-Łojasiewicz or PL inequality), which is written as

$$\langle \nabla f(X), D \rangle = \|\nabla f(X)\|_F^2 \geq \mu(f(X) - f^\star) \quad \text{where } \mu > 0 \text{ and } f^\star = \min_X f(X). \tag{4}$$

Substituting the inequality (4) into the Taylor-like expansion (3) leads to

$$f(X - \alpha D) \leq f(X) - \alpha \|\nabla f(X)\|_F^2 + \alpha^2 (L/2) \|\nabla f(X)\|_F^2$$
$$f(X - \alpha D) - f^\star \leq [1 - \mu\alpha(1 - \alpha L/2)] \cdot (f(X) - f^\star). \tag{5}$$

Here, we can always pick a small enough step-size $\alpha$ to guarantee linear convergence:

$$Q = 1 - \mu\alpha + \mu\alpha^2 L/2 < 1 \implies f(X_k) - f^\star \leq Q^k[f(X_0) - f^\star]. \tag{6}$$

In particular, picking the optimal step-size $\alpha = 1/L$ minimizes the convergence quotient $Q = 1 - 1/(2\kappa)$, where $\kappa = L/\mu$ is the usual *condition number*. This shows that, with an optimal step-size, gradient descent needs at most $O(\kappa \log(1/\epsilon))$ iterations to find an $\epsilon$-suboptimal $X$.

Matrix sensing with exact rank $r = r^\star$ is easily shown to satisfy gradient dominance (4) by manipulating existing results on (restricted) local strong convexity. In the over-parameterized case $r > r^\star$, however, local strong convexity is lost, and gradient dominance can fail to hold. Indeed, consider the following instance of matrix sensing, with true rank $r^\star = 1$, search rank $r = 2$, and $\mathcal{A}$ set to the identity

$$f(X) = \|XX^T - zz^T\|_F^2 \text{ where } X = \begin{bmatrix} 1 & 0 \\ 0 & \xi \end{bmatrix} \text{ and } z = \begin{bmatrix} 1 \\ 0 \end{bmatrix}. \tag{7}$$

We can verify that $\|\nabla f(X)\|^2 = 4\xi^2[f(X) - f^\star]$, and this suggests that $f$ satisfies gradient dominance (4) with a constant of $\mu \leq 2\xi^2$. But $\xi$ is itself a variable that goes to zero as the candidate $XX^T$ approaches to ground truth $zz^T$. For every fixed $\mu > 0$ in the gradient dominance condition (4), we can find a counterexample $X$ in (7) with $\xi < \sqrt{\mu}/2$. Therefore, we must conclude that gradient dominance fails to hold, because the inequality in (4) can only hold for $\mu = 0$.

In fact, this same example also shows why classical gradient descent slows down to a sublinear rate. Applying gradient descent $X_{k+1} = X_k - \alpha\nabla f(X_k)$ with fixed step-size $\alpha$ to (7) yields a sequence of iterates of the same form

$$X_0 = \begin{bmatrix} 1 & 0 \\ 0 & \xi_0 \end{bmatrix}, \qquad X_{k+1} = \begin{bmatrix} 1 & 0 \\ 0 & \xi_{k+1} \end{bmatrix} = \begin{bmatrix} 1 & 0 \\ 0 & \xi_k - \alpha\xi_k^3 \end{bmatrix},$$

from which we can verify that $f(X_{k+1}) = (1 - \alpha\xi_k^2)^4 \cdot f(X_k)$. As each $k$-th $X_k X_k^T$ approaches $zz^T$, the element $\xi_k$ converges towards zero, and the convergence quotient $Q = (1 - \alpha\xi_k^2)^4$ approaches 1. We see a process of diminishing returns: every improvement to $f$ worsens the quotient $Q$, thereby reducing the progress achievable in the subsequent step. This is precisely the notion that characterizes sublinear convergence.

## 5   Linear Convergence for the Noiseless Case

To understand how it is possible make gradient descent converge linearly in the over-parameterized regime, we begin by considering gradient method under a *change of metric*. Let $\mathbf{P}$ be a real symmetric, positive definite $nr \times nr$ matrix. We define a corresponding $P$-inner product, $P$-norm, and dual $P$-norm on $\mathbb{R}^{n \times r}$ as follows

$$\langle X, Y \rangle_P \stackrel{\text{def}}{=} \text{vec}(X)^T \mathbf{P} \text{vec}(Y), \quad \|X\|_P \stackrel{\text{def}}{=} \sqrt{\langle X, X \rangle_P}, \quad \|X\|_{P*} \stackrel{\text{def}}{=} \sqrt{\text{vec}(X)^T \mathbf{P}^{-1} \text{vec}(X)},$$

where $\text{vec} : \mathbb{R}^{n \times r} \to \mathbb{R}^{nr}$ is the usual column-stacking operation. Consider descending in the direction $D$ satisfying $\text{vec}(D) = \mathbf{P}^{-1}\text{vec}(\nabla f(X))$; the resulting decrement in $f$ can be quantified by a $P$-norm analog of the Taylor-like expansion (3)

$$f(X - \alpha D) \leq f(X) - \alpha\langle \nabla f(X), D \rangle + \alpha^2 (L_P/2)\|D\|_P^2 \tag{8}$$

$$= f(X) - \alpha(1 - \alpha(L_P/2))\|\nabla f(X)\|_{P*}^2 \tag{9}$$

where $L_P$ is a $P$-norm gradient Lipschitz constant. If we can demonstrate gradient dominance under the dual $P$-norm,

$$\|\nabla f(X)\|_{P*}^2 \geq \mu_P(f(X) - f^\star) \quad \text{where } \mu_P > 0 \text{ and } f^\star = \min f(X), \tag{10}$$

then we have the desired linear convergence

$$f(X - \alpha D) - f^\star \leq [1 - \mu_P\alpha(1 - \alpha L_P/2)] \cdot (f(X) - f^\star) \tag{11}$$

$$= [1 - 1/(2\kappa_P)] \cdot (f(X) - f^\star) \text{ with } \alpha = 1/L_P, \tag{12}$$

in which the condition number $\kappa_P = L_P/\mu_P$ should be upper-bounded. To make the most progress per iteration, we want to pick a metric $\mathbf{P}$ to make the condition number $\kappa_P$ as small as possible.

The best choice of $\mathbf{P}$ for the fastest convergence is simply the Hessian $\nabla^2 f(X)$ itself, but this simply recovers Newton's method, which would force us to invert a large $nr \times nr$ matrix to compute the search direction $D$ at every iteration. Instead, we look for a *preconditioner* $\mathbf{P}$ that is cheap to apply while still assuring a relatively small condition number $\kappa_P$. The following choice is particularly interesting (the Kronecker product $\otimes$ is defined to satisfy $\mathrm{vec}(AXB^T) = (B \otimes A)\mathrm{vec}(X)$)

$$\mathbf{P} = (X^T X + \eta I_r) \otimes I_n = X^T X \otimes I_n + \eta I_{nr},$$

because the resulting $D = \nabla f(X)(X^T X + \eta I)^{-1}$ allow us to *interpolate* between regular GD and the ScaledGD of Tong et al. [11]. Indeed, we recover regular GD in the limit $\eta \to \infty$, but as we saw in Section 4, gradient dominance (10) fails to hold, so the condition number $\kappa_P = L_P/\mu_P$ grows unbounded as $\mu_P \to 0$. Instead, setting $\eta = 0$ recovers ScaledGD. The key insight of Tong et al. [11] is that under this choice of $\mathbf{P}$, gradient dominance (10) is guaranteed to hold, with a large value of $\mu_P$ that is independent of the current iterate and the ground truth. But as we will now show, this change of metric can magnify the Lipschitz constant $L_P$ by a factor of $\lambda_{\min}^{-1}(X^T X)$, so the condition number $\kappa_P = L_P/\mu_P$ becomes unbounded in the over-parameterized regime.

**Lemma 2** (Lipschitz-like inequality). *Let $\|D\|_P = \|D(X^T X + \eta I_r)^{1/2}\|_F$. Then we have*

$$f(X + D) \le f(X) + \langle \nabla f(X), D \rangle + \frac{1}{2} L_P(X, D) \|D\|_P^2$$

*where*

$$L_P(X, D) = 2(1 + \delta)\left[ 4 + \frac{2\|XX^T - M^\star\|_F + 4\|D\|_P}{\lambda_{\min}(X^T X) + \eta} + \left( \frac{\|D\|_P}{\lambda_{\min}(X^T X) + \eta} \right)^2 \right]$$

**Lemma 3** (Bounded gradient). *For the search direction $D = \nabla f(X)(X^T X + \eta I)^{-1}$, we have $\|D\|_P^2 = \|\nabla f(X)\|_{P*}^2 \le 16(1 + \delta)f(X)$.*

The proofs of Lemma 2 and Lemma 3 follows from straightforward linear algebra, and can be found in the Appendix. Substituting Lemma 3 into Lemma 2, we see for ScaledGD (with $\eta = 0$) that the Lipschitz-like constant is bounded as follows

$$L_P(X, D) \lesssim \left( \|XX^T - M^\star\|_F / \lambda_{\min}(X^T X) \right)^2. \tag{13}$$

In the exact rank case $r = r^\star$, the distance of $X$ from singularity can be lower-bounded, within a "good" neighborhood of the ground truth, since $\lambda_{\min}(X^T X) = \lambda_r(X^T X)$ and

$$\|XX^T - M^\star\|_F \le \rho\lambda_r(M^\star), \quad \rho < 1 \implies \lambda_r(X^T X) \ge (1 - \rho)\lambda_r(M^\star) > 0. \tag{14}$$

Within this "good" neighborhood, substituting (14) into (13) yields a Lipschitz constant $L_P$ that depends only on the radius $\rho$. The resulting iterations converge rapidly, independent of any ill-conditioning in the model $XX^T$ nor in the ground-truth $M^\star$. In turn, ScaledGD can be initialized within the good neighborhood using spectral initialization (see Proposition 6 below).

In the over-parameterized case $r > r^\star$, however, the iterate $X$ must become singular in order for $XX^T$ to converge to $M^\star$, and the radius of the "good" neighborhood reduces to zero. The ScaledGD direction guarantees a large linear progress no matter how singular $X$ may be, but the method may not be able to take a substantial step in this direction if $X$ becomes singular too quickly. To illustrate: the algorithm would fail entirely if it lands at on a point where $\lambda_{\min}(X^T X) = 0$ but $XX^T \ne M^\star$.

While regular GD struggles to make the smallest eigenvalues of $XX^T$ converge to zero, ScaledGD gets in trouble by making these eigenvalues converge quickly. In finding a good mix between these two methods, an intuitive idea is to use the damping parameter $\eta$ to control the rate at which $X$ becomes singular. More rigorously, we can pick an $\eta \approx \|XX^T - ZZ^T\|_F$ and use Lemma 2 to keep the Lipschitz constant $L_P$ bounded. Substituting Lemma 3 into Lemma 2 and using RIP to upper-bound $f(X) \le (1 + \delta)\|XX^T - M^\star\|_F^2$ and $\delta \le 1$ yields

$$\eta \ge C_{\mathrm{lb}}\|XX^T - ZZ^T\|_F \implies L_P(X, D) \le 16 + 136/C_{\mathrm{lb}} + 256/C_{\mathrm{lb}}^2. \tag{15}$$

However, the gradient dominance condition (10) will necessarily fail if $\eta$ is set too large. Our main result in this paper is that keeping $\eta$ within the same order of magnitude as the error norm $\|XX^T - ZZ^T\|_F$ is enough to maintain gradient dominance. The following is the noiseless version of this result.

**Theorem 4** (Noiseless gradient dominance). *Let $\min_X f(X) = 0$ for $M^\star \neq 0$. Suppose that $X$ satisfies $f(X) \leq \rho^2 \cdot (1 - \delta)\lambda_{r^\star}^2(M^\star)$ with radius $\rho > 0$ that satisfies $\rho^2/(1 - \rho^2) \leq (1 - \delta^2)/2$. Then, we have*

$$\eta \leq C_{\text{ub}}\|XX^T - ZZ^T\|_F \implies \|\nabla f(X)\|_{P*}^2 \geq 2\mu_P f(X)$$

*where*

$$\mu_P = \left(\sqrt{\frac{1 + \delta^2}{2}} - \delta\right)^2 \cdot \min\left\{\left(\frac{C_{\text{ub}}}{\sqrt{2} - 1}\right)^{-1}, \left(1 + 3C_{\text{ub}}\sqrt{\frac{(r - r^\star)}{1 - \delta^2}}\right)^{-1}\right\}. \quad (16)$$

The proof of Theorem 4 is involved and we defer the details to the Appendix. In the noiseless case, we get a good estimate of $\eta$ for free as a consequence of RIP:

$$\eta = \sqrt{f(X)} \implies \sqrt{1 - \delta}\|XX^T - M^\star\|_F \leq \eta \leq \sqrt{1 + \delta}\|XX^T - M^\star\|_F.$$

Repeating (8)-(12) with Lemma 2, (15) and (16) yields our main result below.

**Corollary 5** (Linear convergence). *Let $X$ satisfy the same initial conditions as in Theorem 4. The search direction $D = \nabla f(X)(X^T X + \eta I)^{-1}$ with damping parameter $\eta = \sqrt{f(X)}$ and step-size $\alpha \leq 1/L_P$ yields*

$$f(X - \alpha D) \leq (1 - \alpha\mu_P/2) f(X)$$

*where $L_P$ is as in (15) with $C_{\text{lb}} = \sqrt{1 - \delta}$ and $\mu_P$ is as in (16) with $C_{\text{ub}} = \sqrt{1 + \delta}$.*

For a fixed RIP constant $\delta$, Corollary 5 says that PrecGD converges at a linear rate that is independent of the current iterate $X$, and also independent of possible ill-conditioning in the ground truth. However, it does require an initial point $X_0$ that satisfies

$$\|\mathcal{A}(X_0 X_0^T - M^*)\|^2 < \rho^2(1 - \delta)\lambda_{r^*}(M^\star)^2 \quad (17)$$

with a radius $\rho > 0$ satisfying $\rho^2/(1 - \rho^2) \leq (1 - \delta^2)/2$. Such an initial point can be found using spectral initialization, even if the measurements are tainted with noise. Concretely, we choose the initial point $X_0$ as

$$X_0 = \mathcal{P}_r\left(\frac{1}{m}\sum_{i=1}^m y_i A_i\right) \text{ where } \mathcal{P}_r(M) = \arg\min_{X \in \mathbb{R}^{n \times r}}\|XX^T - M\|_F, \quad (18)$$

where we recall that $y = \mathcal{A}(M^\star) + \epsilon$ are the $m$ possibly noisy measurements collected of the ground truth, and that the rank-$r$ projection operator can be efficiently implemented with a singular value decomposition. The proof of the following proposition can be found in the appendix.

**Proposition 6** (Spectral Initialization). *Suppose that $\delta \leq (8\kappa\sqrt{r^*})^{-1}$ and $m \gtrsim \frac{1+\delta}{1-\delta}\frac{\sigma^2 rn\log n}{\rho^2\lambda_{r^\star}^2(M^\star)}$ where $\kappa = \lambda_1(M^\star)/\lambda_{r^\star}(M^\star)$. Then, with high probability, the initial point $X_0$ produced by (18) satisfies the radius condition (17).*

However, if the measurements $y$ are noisy, then $\sqrt{f(X)} = \|\mathcal{A}(XX^T - M^\star) + \varepsilon\|$ now gives a biased estimate of our desired damping parameter $\eta$. In the next section, we show that a good choice of $\eta_k$ is available based on an approximation of the noise variance.

# 6 Extension to Noisy Setting

In this section, we extend our analysis to the matrix sensing with noisy measurements. Our main goal is to show that, with a proper choice of the damping coefficient $\eta$, the proposed algorithm converges linearly to an "optimal" estimation error.

**Theorem 7** (Noisy measurements with optimal $\eta$). *Suppose that the noise vector $\epsilon \in \mathbb{R}^m$ has sub-Gaussian entries with zero mean and variance $\sigma^2 = \frac{1}{m}\sum_{i=1}^m \mathbb{E}[\epsilon_i^2]$. Moreover, suppose that $\eta_k = \frac{1}{\sqrt{m}}\|\mathcal{A}(X_k X_k^T - M^*)\|$, for $k = 0, 1, \ldots, K$, and that the initial point $X_0$ satisfies $\|\mathcal{A}(X_0 X_0^T - M^*)\|^2 < \rho^2(1-\delta)\lambda_{r^\star}(M^\star)^2$. Consider $k^* = \arg\min_k \eta_k$, and suppose that the step-size $\alpha \leq 1/L$, where $L > 0$ is a constant that only depends on $\delta$. Then, with high probability, we have*

$$\|X_{k^*}X_{k^*}^T - M^\star\|_F^2 \lesssim \max\left\{\frac{1 + \delta}{1 - \delta}\left(1 - \alpha\frac{\mu_P}{2}\right)^K\|X_0 X_0^T - M^*\|_F^2, \mathcal{E}_{stat}\right\}, \quad (19)$$

*where $\mathcal{E}_{stat} := \frac{\sigma^2 nr\log n}{\mu_P(1-\delta)m}$.*

Assuming fixed parameters for the problem, the above theorem shows that PrecGD outputs a solution with an estimation error of $\mathcal{O}(\mathcal{E}_{stat})$ in $\mathcal{O}(\log(1/\mathcal{E}_{stat}))$ iterations. Moreover, the error $\mathcal{O}(\mathcal{E}_{stat})$ is minimax optimal (modulo logarithmic factors), and cannot be improved significantly. In particular, Candes and Plan [14] showed that *any* estimator $\widehat{X}$ must satisfy $\|\widehat{X}\widehat{X}^T - M^*\|_F^2 \gtrsim \sigma^2 nr/m$ with non-negligible probability. The classical methods for achieving this minimax rate suffer from computationally-prohibitive per iteration costs [15, 21, 48]. Regular gradient descent alleviates this issue at the expense of a slower convergence rate of $\mathcal{O}(\sqrt{1/\mathcal{E}_{stat}})$ [8]. Our proposed PrecGD achieves the best of both worlds: it converges to the minimax optimal error with cheap per-iteration complexity of $\mathcal{O}(nr^2 + r^3)$, while benefiting from an exponentially faster convergence rate than regular gradient descent in the over-parameterized regime.

Theorem 7 highlights the critical role of the damping coefficient $\eta$ in the guaranteed linear convergence of the algorithm. In the noiseless regime, we showed in the previous section that an "optimal" choice $\eta = \sqrt{f(X)}$ is available for free. In the noisy setting, however, the same choice of $\eta$ becomes biased by the noise variance, and is therefore no longer optimal. As is typically the case for regularized estimation methods [49–51], selecting the ideal parameter would amount to some kind of *resampling*, such as via cross-validation or bootstrapping [52–54], which is generally expensive to implement and use in practice. As an alternative approach, we show in our next theorem that a good choice of $\eta$ is available based on an approximation of the noise variance $\sigma^2$.

**Theorem 8** (Noisy measurements with variance proxy). *Suppose that the noise vector $\epsilon \in \mathbb{R}^m$ has sub-Gaussian entries with zero mean and variance $\sigma^2 = \frac{1}{m}\sum_{i=1}^m \mathbb{E}[\epsilon_i^2]$. Moreover, suppose that $\eta_k = \sqrt{|f(X_k) - \hat{\sigma}^2|}$ for $k = 0, 1, \ldots, K$, where $\hat{\sigma}^2$ is an approximation of $\sigma^2$, and that the initial point $X_0$ satisfies $\|\mathcal{A}(X_0 X_0^T - M^*)\|_F^2 < \rho^2(1-\delta)\lambda_{r^*}(M^*)^2$. Consider $k^* = \arg\min_k \eta_k$, and suppose that the step-size $\alpha \leq 1/L$, where $L > 0$ is a constant that only depends on $\delta$. Then, with high probability, we have*

$$\|X_{k^*}X_{k^*}^T - M^*\|_F^2 \lesssim \max\left\{\frac{1+\delta}{1-\delta}\left(1 - \alpha\frac{\mu_P}{2}\right)^K \|X_0 X_0^T - M^*\|_F^2, \mathcal{E}_{stat}, \mathcal{E}_{dev}, \mathcal{E}_{var}\right\}, \quad (20)$$

*where*

$$\mathcal{E}_{stat} := \frac{\sigma^2 nr \log n}{\mu_P(1-\delta)m}, \quad \mathcal{E}_{dev} := \frac{\sigma^2}{1-\delta}\sqrt{\frac{\log n}{m}}, \quad \mathcal{E}_{var} := |\sigma^2 - \hat{\sigma}^2|. \quad (21)$$

In the above theorem, $\mathcal{E}_{dev}$ captures the deviation of the empirical variance $\frac{1}{m}\sum_{i=1}^m \epsilon_i^2$ from its expectation $\sigma^2$. On the other hand, $\mathcal{E}_{var}$ captures the approximation error of the true variance. According to Theorem 8, it is possible to chose the damping factor $\eta_k$ merely based on $f(X_k)$ and an approximation of $\sigma^2$, at the expense of a suboptimal estimation error rate. In particular, suppose that the noise variance is known precisely, i.e., $\hat{\sigma}^2 = \sigma^2$. Then, the above theorem implies that the estimation error is reduced to

$$\|X_{k^*}X_{k^*}^T - M^*\|_F^2 \lesssim \max\{\mathcal{E}_{stat}, \mathcal{E}_{dev}\} \quad \text{after} \quad \mathcal{O}\left(\log\left(\frac{1}{\max\{\mathcal{E}_{stat}, \mathcal{E}_{dev}\}}\right)\right) \quad \text{iterations.}$$

If $m$ is not too large, i.e., $m \lesssim \sigma^2 n^2 r^2 \log n$, the estimation error can be improved to $\|X_{k^*}X_{k^*}^T - M^*\|_F^2 \lesssim \mathcal{E}_{stat}$, which is again optimal (modulo logarithmic factors). As $m$ increases, the estimation error will become smaller, but the convergence rate will decrease. This suboptimal rate is due to the heavy tail phenomenon arising from the concentration of the noise variance. In particular, one can write

$$f(X) - \sigma^2 = \frac{1}{m}\|\mathcal{A}(XX^T - M^\star)\|^2 + \underbrace{\frac{1}{m}\|\epsilon\|^2 - \sigma^2}_{\text{variance deviation}} + \underbrace{\frac{2}{m}\langle\mathcal{A}(ZZ^T - XX^T), \epsilon\rangle}_{\text{cross-term}} \quad (22)$$

Evidently, $f(X) - \sigma^2$ is in the order of $\frac{1}{m}\|\mathcal{A}(XX^T - M^\star)\|^2$ if both variance deviation and cross-term are dominated by $\frac{1}{m}\|\mathcal{A}(XX^T - M^\star)\|^2$. In the proof of Theorem 8, we show that, with high probability, the variance deviation is upper bounded by $(1-\delta)\mathcal{E}_{dev}$ and it dominates the cross-term. This implies that the choice of $\eta = \sqrt{|f(X) - \sigma^2|}$ behaves similar to $\frac{1}{\sqrt{m}}\|\mathcal{A}(XX^T - M^\star)\|$, and hence, the result of Theorem 7 can be invoked, so long as

$$\frac{1}{m}\|\mathcal{A}(XX^T - M^\star)\|^2 \geq (1-\delta)\|XX^T - M^\star\|_F^2 \gtrsim (1-\delta)\mathcal{E}_{dev}.$$

# 7 Numerical Experiments

Finally, we numerically compare PrecGD on other matrix factorization problems that fall outside of the matrix sensing framework. We consider the $\ell_p$ empirical loss $f_p(X) = \sum_{i=1}^{m} |\langle A_i, XX^T - M^\star \rangle|^p$ for $1 \le p < 2$, in order to gauge the effectiveness of PrecGD for increasing nonsmooth loss functions. Here, we set the damping parameter $\eta_k = [f_p(X_k)]^{1/p}$ as a heuristic for the error $\|XX^T - M^\star\|_F$. The data matrices $A_1, \ldots, A_m$ were taken from [13, Example 12], the ground truth $M^\star = ZZ^T$ was constructed by sampling each column of $Z \in \mathbb{R}^{n \times r^\star}$ from the standard Gaussian, and then rescaling the last column to achieve a desired condition number.

The recent work of Tong et al. [55] showed that in the exactly-parameterized setting, ScaledGD works well for the $\ell_1$ loss function. In particular, if the initial point is close to the ground truth, then with a Polyak stepsize $\alpha_k = f(X_k)/\|\nabla f(X_k)\|_P^*$, ScaledGD converges linearly to the ground truth. However, these theoretical guarantees no longer hold in the over-parameterized regime.

When $r > r^*$, our numerical experiments show that ScaledGD blows up due to singularity near the ground truth while PrecGD continues to converge linearly in this nonsmooth, over-parameterized setting. In Figure 2 we compare GD, ScaledGD and PrecGD in the exact and over-parameterized regimes for the $\ell_p$ norm, with $p = 1.1, 1.4$ and $1.7$. For ScaledGD and PrecGD, we used a modified version of the Polyak step-size where $\alpha_k = f(X_k)^p/\|\nabla f(X_k)\|_P^*$. For GD we use a decaying stepsize. When $r = r^*$, we see that both ScaledGD and PrecGD converge linearly, but GD stagnates due to ill-conditioning of the ground truth. When $r > r^*$, GD still converges slowly and ScaledGD blows up very quickly, while PrecGD continues to converge reliably.

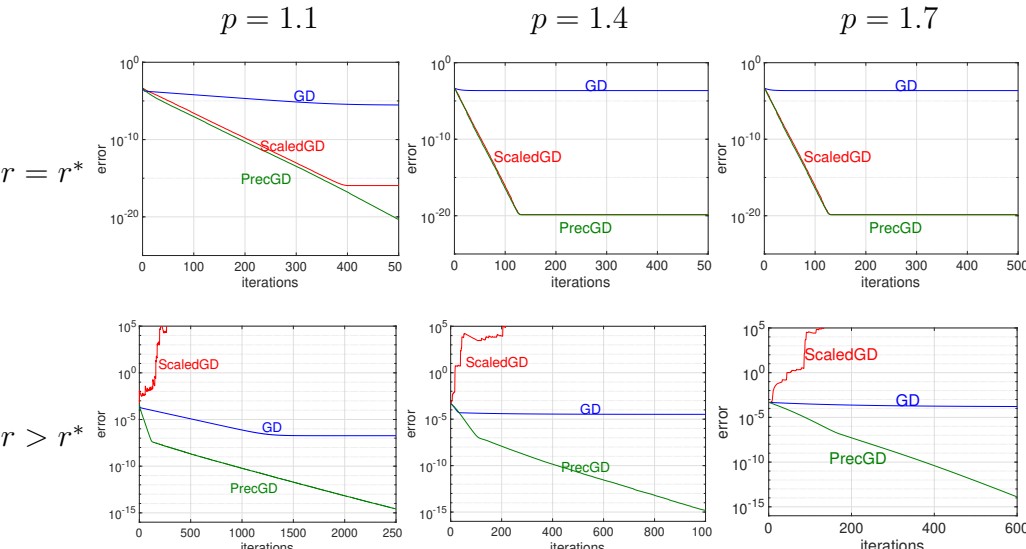

Figure 2: **Nonconvex matrix factorization with the $\ell_p$ empirical loss**. We compare $\ell_p$ matrix sensing with $n = 10$ and $r^\star = 2$ and $\mathcal{A}$ taken from [13]. The ground truth is chosen to be ill-conditioned ($\kappa = 10^2$). For ScaledGD and PrecGD, we use the Polyak step-size in [55]. For GD we use a decaying step-size. (**Top** $r = r^*$) For all three values of $p$, GD stagnates due to the ill-conditioning of the ground truth, while ScaledGD and PrecGD converge linearly in all three cases. (**Bottom** $r > r^*$) With $r = 4$, the problem is over-parameterized. GD again converges slowly and ScaledGD is sporadic due to near-singularity caused by over-parameterization. Once again we see PrecGD converge at a linear rate.

# 8 Conclusions

In this paper, we propose a *preconditioned* gradient descent or PrecGD for nonconvex matrix factorization with a comparable per-iteration cost to classical gradient descent. For over-parameterized matrix sensing, gradient descent slows down to a sublinear convergence rate, but PrecGD restores

the convergence rate back to linear, while also making the iterations immune to ill-conditioning in the ground truth. While the thoeretical analysis in our paper uses some properties specific to RIP matrix sensing, our numerical experiments find that PrecGD works well for even for nonsmooth loss functions. We believe that these current results can be extended to similar problems such as matrix completion and robust PCA, where properties like incoherence can be used to select the damping parameter $\eta_k$ with the desired properties, so that PrecGD converges linearly as well. It remains future work to provide rigorous justification for these observations.

## Acknowledgements

G.Z. and R.Y.Z are supported by the NSF CAREER Award ECCS-2047462. S.F. is supported by MICDE Catalyst Grant and MIDAS PODS Grant. We also thank an anonymous reviewer who provided a simplified proof of Lemma 14 and made various insightful comments to help us improve an earlier version of this work.

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
