# A    Preliminaries for the Noiseless Case

Recall that the matrix inner product is defined $\langle X, Y \rangle \overset{\text{def}}{=} \text{Tr}\left(X^T Y\right)$, and that it induces the Frobenius norm as $\|X\|_F = \sqrt{\langle X, X \rangle}$. The vectorization $\text{vec}(X)$ is the usual column-stacking operation that turns an $m \times n$ matrix into a length-$mn$ vector; it preserves the matrix inner product $\langle X, Y \rangle = \text{vec}(X)^T \text{vec}(Y)$ and the Frobenius norm $\|\text{vec}(X)\| = \|X\|_F$. The Kronecker product $\otimes$ is implicitly defined to satisfy $\text{vec}(AXB^T) = (B \otimes A)\text{vec}X$.

We denote $\lambda_i(M)$ and $\sigma_i(M)$ as the $i$-th eigenvalue and singular value of a symmetric matrix $M = M^T$, ordered from the most positive to the most negative. We will often write $\lambda_{\max}(M)$ and $\lambda_{\min}(M)$ to index the most positive and most negative eigenvalues, and $\sigma_{\max}(M)$ and $\sigma_{\min}(M)$ for the largest and smallest singular values.

We denote $\mathbf{A} = [\text{vec}(A_1), \dots, \text{vec}(A_m)]^T$ as the matrix representation of $\mathcal{A}$, and note that $\mathcal{A}(X) = \mathbf{A}\,\text{vec}(X)$. For fixed $X$ and $M^\star$, we can rewrite $f$ in terms of the error matrix $E$ or its vectorization $\mathbf{e}$ as follows

$$f(X) = \|\mathcal{A}(E)\|^2 = \|\mathbf{A}\mathbf{e}\|^2 \text{ where } E = XX^T - M^\star, \quad \mathbf{e} = \text{vec}(E). \tag{23}$$

The gradient satisfies for any matrix $D \in \mathbb{R}^{n \times r}$

$$\langle \nabla f(X), D \rangle = 2 \left\langle \mathcal{A}\left(XD^T + DX^T\right), \mathcal{A}\left(E\right) \right\rangle. \tag{24}$$

Letting $\mathbf{J}$ denote the Jacobian of the vectorized error $\mathbf{e}$ with respect to $X$ implicitly as the matrix that satisfies

$$\mathbf{J}\,\text{vec}(Y) = \text{vec}(XY^T + YX^T) \qquad \text{for all } Y \in \mathbb{R}^{n \times r}. \tag{25}$$

allows us to write the gradient exactly as $\text{vec}(\nabla f(X)) = 2\mathbf{J}^T \mathbf{A}^T \mathbf{A}\mathbf{e}$. The noisy versions of (23) and (24) are obvious, though we will defer these to Section E.

Recall that $\mathcal{A}$ is assumed to satisfy RIP (Definition 1) with parameters $(2r, \delta)$. Here, we set $m = 1$ without loss of generality to avoid carrying the normalizing constant; the resulting RIP inequality reads

$$(1 - \delta)\|M\|_F^2 \le \|\mathcal{A}(M)\|^2 \le (1 + \delta)\|M\|_F^2 \text{ for all } M \text{ such that } \text{rank}(M) \le 2r, \tag{26}$$

where we recall that $0 \le \delta < 1$. It is easy to see that RIP preserves the Cauchy–Schwarz identity for all rank-$2r$ matrices $G$ and $H$:

$$\langle \mathcal{A}(G), \mathcal{A}(H) \rangle \le \|\mathcal{A}(G)\|\|\mathcal{A}(H)\| \le (1 + \delta)\|G\|_F\|H\|_F. \tag{27}$$

As before, we introduce the preconditioner matrix $P$ as

$$P \overset{\text{def}}{=} X^T X + \eta I_r, \qquad\qquad \mathbf{P} \overset{\text{def}}{=} P \otimes I_n = (X^T X + \eta I_r) \otimes I_n$$

and define a corresponding $P$-inner product, $P$-norm, and dual $P$-norm on $\mathbb{R}^{n \times r}$ as follows

$$\langle X, Y \rangle_P \overset{\text{def}}{=} \text{vec}(X)^T \mathbf{P}\text{vec}(Y) = \left\langle XP^{1/2}, YP^{1/2} \right\rangle = \text{Tr}\left(XPY^T\right), \tag{28a}$$

$$\|X\|_P \overset{\text{def}}{=} \sqrt{\langle X, X \rangle_P} = \|\mathbf{P}^{1/2}\text{vec}(X)\| = \|XP^{1/2}\|_F, \tag{28b}$$

$$\|X\|_{P*} \overset{\text{def}}{=} \max_{\|Y\|_P = 1} \langle Y, X \rangle = \|\mathbf{P}^{-1/2}\text{vec}(X)\| = \|XP^{-1/2}\|_F. \tag{28c}$$

Finally, we will sometimes need to factorize the ground truth $M^\star = ZZ^T$ in terms of the low-rank factor $Z \in \mathbb{R}^{n \times r^\star}$.

# B    Proof of Lipschitz-like Inequality (Lemma 2)

In this section we give a proof of Lemma 2, which is a Lipschitz-like inequality under the $P$-norm. Recall that we proved linear convergence for PrecGD by lower-bounding the linear progress $\langle \nabla f(X), D \rangle$ and upper-bounding $\|D\|_P$.

**Lemma 9** (Lipschitz-like inequality; Lemma 2 restated). *Let* $\|D\|_P = \|D(X^TX + \eta I)^{1/2}\|_F$. *Then we have*

$$f(X + D) \le f(X) + \langle \nabla f(X), D \rangle + \frac{1}{2} L_P(X, D) \|D\|_P^2$$

*where*

$$L_P(X, D) = 2(1 + \delta) \left[ 4 + \frac{2\|XX^T - M^\star\|_F + 4\|D\|_P}{\lambda_{\min}(X^TX) + \eta} + \left( \frac{\|D\|_P}{\lambda_{\min}(X^TX) + \eta} \right)^2 \right]$$

*Proof.* Recall that $E = XX^T - M^\star$. We obtain a Taylor expansion of the quartic polynomial $f$ by directly expanding the quadratic terms

$$f(X + D) = \|\mathcal{A}((X + D)(X + D)^T - M^\star)\|^2$$
$$= \underbrace{\|\mathcal{A}(E)\|^2 + 2\langle \mathcal{A}(E), \mathcal{A}(XD^T + DX^T) \rangle}_{f(X) + \langle \nabla f(X), D \rangle} + \underbrace{2\langle \mathcal{A}(E), \mathcal{A}(DD^T) \rangle + \|\mathcal{A}(XD^T + DX^T)\|^2}_{\frac{1}{2}\langle \nabla^2 f(X)[D], D \rangle}$$
$$+ \underbrace{2\langle \mathcal{A}(XD^T + DX^T), \mathcal{A}(DD^T) \rangle}_{\frac{1}{6}\langle \nabla^3 f(X)[D, D], D \rangle} + \underbrace{\|\mathcal{A}(DD^T)\|^2}_{\frac{1}{24}\langle \nabla^4 f(X)[D, D, D], D \rangle} .$$

We evoke RIP to preserve Cauchy–Schwarz as in (27), and then bound the second, third, and fourth order terms

$$T = 2\langle \mathcal{A}(E), \mathcal{A}(DD^T) \rangle + \|\mathcal{A}(XD^T + DX^T)\|^2 + 2\langle \mathcal{A}(XD^T + DX^T), \mathcal{A}(DD^T) \rangle + \|\mathcal{A}(DD^T)\|^2$$
$$\le (1 + \delta) \left( 2\|E\|_F \|DD^T\|_F + \|XD^T + DX^T\|^2 + 2\|XD^T + DX^T\|_F \|DD^T\|_F + \|DD^T\|_F^2 \right)$$
$$\le (1 + \delta) \left( 2\|E\|_F \|D\|_F^2 + 4\|XD^T\|^2 + 4\|XD^T\|_F \|D\|_F^2 + \|D\|_F^4 \right) \quad (29)$$

where the third line uses $\|DD^T\|_F \le \|D\|_F^2$ and $\|XD^T + DX^T\|_F \le 2\|XD^T\|_F$. Now, write $d = \text{vec}(D)$ and observe that

$$\|D\|_F^2 = d^T d = (d^T \mathbf{P}^{1/2}) \mathbf{P}^{-1} (\mathbf{P}^{1/2} d) \le (d^T \mathbf{P} d) \lambda_{\max}(\mathbf{P}^{-1}) = \|D\|_P^2 / \lambda_{\min}(\mathbf{P}). \quad (30)$$

Similarly, we have

$$\|XD^T\|_F = \|XP^{-1/2}P^{1/2}D^T\|_F \le \sigma_{\max}(XP^{-1/2}) \|P^{1/2}D^T\|_F \le \|D\|_P. \quad (31)$$

The final inequality uses $\|P^{1/2}D^T\|_F = \|DP^{1/2}\|_F = \|D\|_P$ and that

$$\sigma_{\max}(XP^{-1/2}) = \sigma_{\max}[X(X^TX + \eta I)^{-1/2}] = \sigma_{\max}(X)/\sqrt{\sigma_{\max}^2(X) + \eta} \le 1. \quad (32)$$

Substituting (30) and (31) into (29) yields

$$T \le (1 + \delta) \left( 2\|E\|_F \frac{\|D\|_P^2}{\lambda_{\min}(\mathbf{P})} + 4\|D\|_P^2 + \frac{4\|D\|_P^3}{\lambda_{\min}(\mathbf{P})} + \frac{\|D\|_P^4}{\lambda_{\min}^2(\mathbf{P})} \right) = \frac{1}{2} L_P(X, D)\|D\|_P^2$$

where we substitute $\lambda_{\min}(\mathbf{P}) = \lambda_{\min}(X^TX) + \eta$. $\qquad \square$

## C   Proof of Bounded Gradient (Lemma 3)

In this section we prove Lemma 3, which shows that the gradient measured in the dual $P$-norm $\|\nabla f(X)\|_{P*}$ is controlled by the objective value as $\sqrt{f(X)}$.

**Lemma 10** (Bounded Gradient; Lemma 3 restated). *For the search direction $D = \nabla f(X)(X^TX + \eta I)^{-1}$, we have $\|D\|_P^2 = \|\nabla f(X)\|_{P*}^2 \le 16(1 + \delta)f(X)$.*

*Proof.* We apply the variation definition of the dual $P$-norm in (28c) to the gradient in (24) to obtain

$$\|\nabla f(X)\|_{P*} = \max_{\|Y\|_P = 1} \langle \nabla f(X), Y \rangle = \max_{\|Y\|_P = 1} 2 \langle \mathcal{A}(XY^T + YX^T), \mathcal{A}(E) \rangle$$
$$\overset{(a)}{\le} 2\|\mathcal{A}(E)\| \max_{\|Y\|_P = 1} \|\mathcal{A}(XY^T + YX^T)\| \overset{(b)}{\le} 4\sqrt{(1 + \delta)f(X)} \max_{\|Y\|_P = 1} \|XY^T\|_F$$

Here (a) applies Cauchy–Schwarz; and (b) substitutes $f(X) = \|\mathcal{A}(E)\|^2$ and $\|\mathcal{A}(M)\| \leq \sqrt{1+\delta}\|M\|_F$ for rank-$2r$ matrix $M$ and $\|XY^T + YX^T\|_F \leq 2\|XY^T\|_F$. Now, we bound the final term

$$\max_{\|Y\|_P=1} \|XY^T\|_F = \max_{\|YP^{1/2}\|_F=1} \|XY^T\|_F = \max_{\|\tilde{Y}\|_F=1} \|XP^{-1/2}\tilde{Y}^T\|_F = \sigma_{\max}(XP^{-1/2}) \leq 1$$

where the final inequality uses (32). $\qquad\square$

# D    Proof of Gradient Dominance (Theorem 4)

In this section we prove our first main result: the gradient $\nabla f(X)$ satisfies gradient dominance the $P$-norm. This is the key insight that allowed us to establish the linear convergence rate of PrecGD in the main text. The theorem is restated below.

**Theorem 11** (Gradient Dominance; Theorem 4 restated). *Let $\min_X f(X) = 0$ for $M^\star \neq 0$. Suppose that $X$ satisfies $f(X) \leq \rho^2 \cdot (1-\delta)\lambda_{r^\star}^2(M^\star)$ with radius $\rho > 0$ that satisfies $\rho^2/(1-\rho^2) \leq (1-\delta^2)/2$. Then, we have*

$$\eta \leq C_{\text{ub}}\|XX^T - M^\star\|_F \quad \Longrightarrow \quad \|\nabla f(X)\|_{P*}^2 \geq \mu_P f(X)$$

*where*

$$\mu_P = \left(\sqrt{\frac{1+\delta^2}{2}} - \delta\right)^2 \cdot \min\left\{\left(1 + \frac{C_{\text{ub}}}{\sqrt{2}-1}\right)^{-1}, \left(1 + 3C_{\text{ub}}\sqrt{\frac{(r-r^\star)}{1-\delta^2}}\right)^{-1}\right\}. \tag{33}$$

The theorem is a consequence of the following lemma, which shows that the PL constant $\mu_P > 0$ is driven in part by the alignment between the model $XX^T$ and the ground truth $M^\star$, and in part in the relationship between $\eta$ and the singular values of $X$. We defer its proof to Section D.1 and first use it to prove Theorem 4.

**Lemma 12** (Gradient lower bound). *Let $XX^T = U\Lambda U^T$ where $\Lambda = \text{diag}(\lambda_1, \ldots, \lambda_r)$, $\lambda_1 \geq \cdots \geq \lambda_r \geq 0$, and $U^T U = I_r$ denote the usual eigenvalue decomposition. Let $U_k$ denote the first $k$ columns of $U$. Then, we have*

$$\|\nabla f(X)\|_{P*}^2 \geq \max_{k \in \{1,2,\ldots,r\}} \frac{2(\cos\theta_k - \delta)^2}{1 + \eta/\lambda_k}\|XX^T - M^\star\|_F^2 \tag{34}$$

*where each $\theta_k$ is defined*

$$\sin\theta_k = \frac{\left\|\left(I - U_k U_k^T\right)\left(XX^T - M^\star\right)\left(I - U_k U_k^T\right)\right\|_F}{\|XX^T - M^\star\|_F}. \tag{35}$$

From Lemma 12, we see that deriving a PL constant $\mu_P$ requires balancing two goals: (1) ensuring that $\cos\theta_k$ is large with respect to the RIP constant $\delta$; (2) ensuring that $\lambda_k(X^T X)$ is large with respect to the damping parameter $\eta$.

As we will soon show, in the case that $k = r$, the corresponding $\cos\theta_r$ is guaranteed to be large with respect to $\delta$, once $XX^T$ converges towards $M^\star$. At the same time, we have by Weyl's inequality

$$\lambda_k(X^T X) = \lambda_k(XX^T) \geq \lambda_k(M^\star) - \|XX^T - M^\star\|_F \text{ for all } k \in \{1, 2, \ldots, r\}.$$

Therefore, when $k = r^\star$ and $XX^T$ is close to $M^\star$, the corresponding $\lambda_{r^\star}(X^T X)$ is guaranteed to be large with respect to $\eta$. However, in order to use Lemma 12 to derive a PL constant $\mu_P > 0$, we actually need $\cos\theta_k$ and $\lambda_k(X^T X)$ to both be large for the *same* value of $k$. It turns out that when $\eta \gtrsim \|XX^T - M^\star\|_F$, it is possible to prove this claim using an inductive argument.

Before we present the complete argument and prove Theorem 4, we state one more lemma that will be used in the proof.

**Lemma 13** (Basis alignment). *Define the $n \times k$ matrix $U_k$ in terms of the first $k$ eigenvectors of $X$ as in Lemma 12. Let $Z \in \mathbb{R}^{n \times r^\star}$ satisfy $\lambda_{\min}(Z^T Z) > 0$ and suppose that $\|XX^T - ZZ^T\|_F \leq \rho\lambda_{\min}(Z^T Z)$ with $\rho \leq 1/\sqrt{2}$. Then,*

$$\frac{\|Z^T(I - U_k U_k^T)Z\|_F}{\|XX^T - ZZ^T\|_F} \leq \frac{1}{\sqrt{2}}\frac{\rho}{\sqrt{1-\rho^2}} \quad \text{for all } k \geq r^\star. \tag{36}$$

Essentially, this lemma states that as the rank-$r$ matrix $XX^T$ converges to the rank-$r^\star$ matrix $M^\star$, the top $r^\star$ eigenvectors of $XX^T$ must necessarily rotate into alignment with $M^\star$. In fact, this is easily verified to be sharp by considering the $r = r^\star = 1$ case; we defer its proof to Section D.2.

With Lemma 12 and Lemma 13, we are ready to prove Theorem 4.

*Proof of Theorem 4.* We pick some $\mu$ satisfying $\delta < \mu < 1$ and prove that $\frac{\rho^2}{1-\rho^2} \leq 1 - \mu^2$ implies $\|\nabla f(X)\|_{P*}^2 \geq \mu_P f(X)$ where

$$\mu_P = (\mu - \delta)^2 \cdot \min\left\{ \left(1 + \frac{C_{\mathrm{ub}}}{\sqrt{2} - 1}\right)^{-1}, \left(1 + 3C_{\mathrm{ub}}\sqrt{\frac{r - r^\star}{1 - \mu^2}}\right)^{-1} \right\}. \tag{37}$$

Then, setting $1 - \mu^2 = \frac{1}{2}(1 - \delta^2)$ yields our desired claim.

To begin, note that the hypothesis $\frac{\rho^2}{1-\rho^2} \leq 1 - \mu^2 \leq 1$ implies $\rho \leq 1/\sqrt{2}$. Denote $E = XX^T - M^\star$. We have

$$\frac{\|\nabla f(X)\|_{P*}^2}{f(X)} \overset{(a)}{\geq} \frac{\|\nabla f(X)\|_{P*}^2}{(1+\delta)\|E\|_F^2} \overset{(b)}{\geq} \frac{2(\cos\theta_k - \delta)^2}{(1+\delta)(1 + \eta/\lambda_k(X^TX))} \overset{(c)}{\geq} \frac{(\cos\theta_k - \delta)^2}{1 + \eta/\lambda_k(X^TX)} \quad \text{for all } k \geq r^\star. \tag{38}$$

Step (a) follows from RIP; Step (b) applies Lemma 12; Step (c) applies $1 + \delta \leq 2$. Equation (38) proves gradient dominance if we can show that both $\lambda_k(X^TX)$ and $\cos\theta_k$ are large for the same $k$. We begin with $k = r^\star$. Here we have by RIP and by hypothesis

$$(1 - \delta)\|XX^T - M^\star\|_F^2 \leq f(X) \leq \rho^2 \cdot (1 - \delta)\lambda_{\min}^2(Z^TZ), \tag{39}$$

which by Weyl's inequality yields

$$\lambda_{r^\star}(X^TX) = \lambda_{r^\star}(XX^T) \geq \lambda_{r^\star}(M^\star) - \|XX^T - M^\star\|_F \geq (1 - \rho)\lambda_{r^\star}(M^\star).$$

This, combined with (39) and our hypothesis $\eta \leq C_{\mathrm{ub}}\|XX^T - ZZ^T\|_F$ and $\rho \leq 1/\sqrt{2}$ gives

$$\frac{\eta}{\lambda_{r^\star}(X^TX)} \leq \frac{\rho C_{\mathrm{ub}}\lambda_{r^\star}(M^\star)}{(1-\rho)\lambda_{r^\star}(M^\star)} = \frac{\rho C_{\mathrm{ub}}}{1 - \rho} \leq \frac{C_{\mathrm{ub}}}{\sqrt{2} - 1}, \tag{40}$$

which shows that $\lambda_{r^\star}(X^TX)$ is large. If $\cos\theta_k \geq \mu$ is also large, then substituting (40) into (38) yields gradient dominance

$$\frac{\|\nabla f(X)\|_{P*}^2}{f(X)} \geq (\mu - \delta)^2 \left(1 + \frac{C_{\mathrm{ub}}}{\sqrt{2} - 1}\right)^{-1},$$

and this yields the first term in (37). If $\cos\theta_k < \mu$ is actually small, then $\sin^2\theta_k > 1 - \mu^2$ is large. We will show that this lower bound on $\sin\theta_k$ actually implies that $\lambda_{k+1}(X^TX)$ will be large.

To see this, let us write $XX^T = U_k\Lambda_k U_k^T + R$ where the $n \times k$ matrix of eigenvectors $U_k$ is defined as in Lemma 12, $\Lambda_k$ is the corresponding $k \times k$ diagonal matrix of eigenvalues, and $U_k^T R = 0$. Denote $\Pi_k = I - U_k U_k^T$ and note that

$$\|\Pi_k(XX^T - M^\star)\Pi_k\|_F = \|\Pi_k XX^T\Pi_k - \Pi_k M^\star\Pi_k\|_F = \|R - \Pi_k M^\star\Pi_k\|_F.$$

By the subadditivity of the norm $\|R - \Pi_k M^\star\Pi_k\|_F \leq \|R\|_F + \|\Pi_k M^\star\Pi_k\|_F$. Dividing both sides by $\|E\|_F$ yields

$$\sin\theta_k = \frac{\|R - \Pi_k M^\star\Pi_k\|_F}{\|E\|_F} \leq \frac{\|\Pi_k M^\star\Pi_k\|_F}{\|E\|_F} + \frac{\|R\|_F}{\|E\|_F}.$$

Since $\rho \leq 1/\sqrt{2}$ by assumption, Lemma 13 yields

$$\frac{\|\Pi_k M^\star\Pi_k\|_F}{\|E\|_F} \leq \frac{1}{\sqrt{2}}\frac{\rho}{\sqrt{1 - \rho^2}} \leq \rho.$$

In addition,

$$\|R\|_F \leq \|R\| \cdot \sqrt{\mathrm{rank}(R)} = \lambda_{k+1}(XX^T) \cdot \sqrt{r - k}.$$

Combining the two inequalities above we get

$$\sqrt{1-\mu^2} \leq \sin\theta_k \leq \frac{1}{\sqrt{2}}\frac{\rho}{\sqrt{1-\rho^2}} + \sqrt{r-k}\cdot\frac{\lambda_{k+1}\left(X^TX\right)}{\|E\|_F}.$$

Rearranging, we get

$$\frac{\lambda_{k+1}\left(X^TX\right)}{\|E\|_F} \geq \frac{1}{\sqrt{r-k}}\left(\sqrt{1-\mu^2} - \frac{1}{\sqrt{2}}\frac{\rho}{\sqrt{1-\rho^2}}\right) \geq \left(1-\frac{1}{\sqrt{2}}\right)\sqrt{\frac{1-\mu^2}{r-k}}.$$

Note that the last inequality above follows from the assumption that $\frac{\rho^2}{1-\rho^2} \leq 1-\mu^2$. Now substituting $\eta \leq C_{\mathrm{ub}}\|XX^T - M^\star\|_F$ and $r-k \leq r-r^\star$ and noting that $\left(1-\frac{1}{\sqrt{2}}\right) \leq 1/3$ we get

$$\frac{\eta}{\lambda_{k+1}(X^TX)} \leq C_{\mathrm{ub}}\frac{\|XX^T - M^\star\|_F}{\lambda_{k+1}(X^TX)} \leq 3C_{\mathrm{ub}}\sqrt{\frac{r-k}{1-\mu^2}} \leq 3C_{\mathrm{ub}}\sqrt{\frac{r-r^\star}{1-\mu^2}}, \qquad (41)$$

which shows that $\lambda_{k+1}(X^TX)$ is large.

If $\cos\theta_{k+1} \geq \mu$ is also large, then substituting (41) into (38) yields gradient dominance

$$\frac{\|\nabla f(X)\|_{P^*}^2}{f(X)} \geq \frac{(\cos\theta_{k+1} - \delta)^2}{1 + \eta/\lambda_{k+1}^2(X)} \geq (\mu-\delta)^2\left(1 + 3C_{\mathrm{ub}}\sqrt{\frac{r-r^\star}{1-\mu^2}}\right)^{-1}, \qquad (42)$$

and this yields the second term in (37) so we are done. If $\cos\theta_{k+1} < \mu$ then we can simply repeat the argument above to show that $\lambda_{k+1}(X^TX)$ is large. We can repeat this process until $k+1 = r$. At this point, we have

$$\cos^2\theta_r = 1 - \sin^2\theta_r \geq 1 - \frac{1}{2}\frac{\rho^2}{1-\rho^2} \geq \mu^2$$

where we used our hypothesis $1-\mu^2 \geq \frac{\rho^2}{1-\rho^2} \geq \frac{1}{2}\frac{\rho^2}{1-\rho^2}$, and substituting (41) into (38) again yields gradient dominance in (42). $\qquad\square$

### D.1 Proof of Gradient Lower Bound (Lemma 12)

In this section we prove Lemma 12, where we prove gradient dominance $\|\nabla f(X)\|_{P^*}^2 \geq \mu_P f(X)$ with a PL constant $\mu_P$ that is proportional to $\cos\theta_k - \delta$ and to $\lambda_k(X^TX)/\eta$. We first prove the following result which will be useful in the proof of Lemma 12.

**Lemma 14.** *Let $\mathcal{A}$ satisfy RIP with parameters $(\zeta, \delta)$, where $\zeta = \mathrm{rank}([X, Z])$. Then, we have*

$$\|\nabla f(X)\|_{P^*} \geq \max_{\|Y\|_P \leq 1}\langle XY^T + YX^T, E\rangle - \delta\|XY^T + YX^T\|_F\|E\|_F \qquad (43)$$

*Proof.* Let $Y$ maximize the right-hand side of (43) and let $W$ be the matrix corrresponding to the orthogonal projection onto $\mathrm{range}(X) + \mathrm{range}(Y)$. Set $\tilde{Y} = WY$, then

$$\langle X\tilde{Y}^T + \tilde{Y}X^T, E\rangle = \langle XY^T, EW\rangle + \langle YX^T, WE\rangle = \langle XY^T + YX^T, E\rangle.$$

On the other hand, we have

$$\|X\tilde{Y}^T + \tilde{Y}X^T\|_F = \|W\left(XY^T + YX^T\right)W\|_F \leq \|XY^T + YX^T\|_F$$

and

$$\|\tilde{Y}\|_P = \|WYP^{1/2}\|_F \leq \|YP^{1/2}\|_F = \|Y\|_P.$$

This means that $\tilde{Y}$ is feasible and makes the right-hand side at least as large as $Y$. Since $Y$ is the maximizer by definition, we conclude that $\tilde{Y}$ also maximizes the right-hand side of (43).

By definition, $\mathrm{range}(\tilde{Y}) \subset \mathrm{range}(X) + \mathrm{range}(Z)$, so $(2r, \delta)$-RIP implies

$$|\langle A(X\tilde{Y}^T + \tilde{Y}X^T), A(E)\rangle - \langle X\tilde{Y}^T + \tilde{Y}X^T, E\rangle| \leq \delta\|X\tilde{Y}^T + \tilde{Y}X^T\|_F\|E\|_F.$$

Now we have

$$\begin{aligned}
\|\nabla f(X)\|_{P*} &= \max_{\|Y\|_P \leq 1} \langle \mathcal{A}(XY^T + YX^T), \mathcal{A}(E) \rangle \\
&\geq \langle \mathcal{A}(X\tilde{Y}^T + \tilde{Y}X^T), \mathcal{A}(E) \rangle \\
&\geq \langle X\tilde{Y}^T + \tilde{Y}X^T, E \rangle - \delta \|X\tilde{Y}^T + \tilde{Y}X^T\|_F \|E\|_F \\
&= \max_{\|Y\|_P \leq 1} \langle XY^T + YX^T, E \rangle - \delta \|XY^T + YX^T\|_F \|E\|_F.
\end{aligned}$$

This completes the proof. $\qquad \square$

*Proof of Lemma 12.* Let $X = \sum_{i=1}^r \sigma_i u_i v_i^T$ with $\|u_i\| = \|v_i\| = 1$ and $\sigma_1 \geq \cdots \geq \sigma_r$ denote the usual singular value decomposition. Observe that the preconditioned Jacobian $\mathbf{J}\mathbf{P}^{-1/2}$ satisfies

$$\mathbf{J}\mathbf{P}^{-1/2}\mathrm{vec}(Y) = \mathrm{vec}(XP^{-1/2}Y^T + YP^{-1/2}X^T) = \mathrm{vec}\left( \sum_{i=1}^r \frac{u_i y_i^T + y_i u_i^T}{\sqrt{1 + \eta/\sigma_i^2}} \right)$$

where $y_i = Yv_i$. This motivates the following family of singular value decompositions

$$\mathbf{U}_k \boldsymbol{\Sigma}_k \mathbf{V}_k^T \mathrm{vec}(Y) = \mathrm{vec}\left( \sum_{i=1}^k \frac{u_i y_i^T + y_i u_i^T}{\sqrt{1 + \eta/\sigma_i^2}} \right) \text{ for all } k \in \{1, 2, \ldots, r\}, \quad \mathbf{J}\mathbf{P}^{-1/2} = \mathbf{U}_r \boldsymbol{\Sigma}_r \mathbf{V}_r^T. \tag{44}$$

Here, the $n^2 \times \zeta_k$ matrix $\mathbf{U}_k$ and the $nr \times \zeta_k$ matrix $\mathbf{V}_k$ have orthonormal columns, and the rank can be verified as $\zeta_k = nk - k(k-1)/2 < nr \leq n^2$. Now, we rewrite Lemma 14 by vectorizing $y = \mathrm{vec}(Y)$ and writing

$$\begin{aligned}
\|\nabla f(X)\|_{P*} &\geq \max_{\|\mathbf{P}^{1/2}y\| \leq 1} \left( \frac{\mathbf{e}^T \mathbf{J} y}{\|\mathbf{e}\| \|\mathbf{J}y\|} - \delta \right) \|\mathbf{e}\| \|\mathbf{J}y\| \stackrel{(a)}{=} \max_{\|y'\| \leq 1} \left( \frac{\mathbf{e}^T \mathbf{J}\mathbf{P}^{-1/2}y}{\|\mathbf{e}\| \|\mathbf{J}\mathbf{P}^{-1/2}y\|} - \delta \right) \|\mathbf{e}\| \|\mathbf{J}\mathbf{P}^{-1/2}y\| \\
&\stackrel{(b)}{=} \max_{\|y'\| \leq 1} \left( \frac{\mathbf{e}^T \mathbf{U}_r \boldsymbol{\Sigma}_r \mathbf{V}_r^T y}{\|\mathbf{e}\| \|\mathbf{U}_r \boldsymbol{\Sigma}_r \mathbf{V}_r^T y\|} - \delta \right) \|\mathbf{e}\| \|\mathbf{U}_r \boldsymbol{\Sigma}_r \mathbf{V}_r^T y\| \\
&\stackrel{(c)}{\geq} \left( \frac{\mathbf{e}^T \mathbf{U}_k \mathbf{U}_k^T \mathbf{e}}{\|\mathbf{e}\| \|\mathbf{U}_k^T \mathbf{e}\|} - \delta \right) \|\mathbf{e}\| \frac{\|\mathbf{U}_k^T \mathbf{e}\|}{\|\boldsymbol{\Sigma}_k^{-1} \mathbf{U}_k^T \mathbf{e}\|} \stackrel{(d)}{\geq} \left( \frac{\|\mathbf{U}_k^T \mathbf{e}\|}{\|\mathbf{e}\|} - \delta \right) \|\mathbf{e}\| \lambda_{\min}(\boldsymbol{\Sigma}_k).
\end{aligned}$$

Step (a) makes a change of variables $y \leftarrow \mathbf{P}^{1/2}y$; Step (b) substitutes (44); Step (c) substitutes the heuristic choice $y = d/\|d\|$ where $d = \mathbf{V}_k \boldsymbol{\Sigma}_k^{-1} \mathbf{U}_k^T \mathbf{e}$; Step (d) notes that $\mathbf{e}^T \mathbf{U}_k \mathbf{U}_k^T \mathbf{e} = \|\mathbf{U}_k^T \mathbf{e}\|^2$ and that $\|\boldsymbol{\Sigma}_k^{-1} \mathbf{U}_k^T \mathbf{e}\| \leq \|\mathbf{U}_k^T \mathbf{e}\| \cdot \lambda_{\max}(\boldsymbol{\Sigma}_k^{-1}) = \|\mathbf{U}_k^T \mathbf{e}\|/\lambda_{\min}(\boldsymbol{\Sigma}_k)$. Finally, we can mechanically verify from (44) that

$$\cos^2 \theta_k \stackrel{\mathrm{def}}{=} \frac{\|\mathbf{U}_k^T \mathbf{e}\|^2}{\|\mathbf{e}\|^2} = 1 - \frac{\|(I - \mathbf{U}_k^T \mathbf{U}_k)\mathbf{e}\|^2}{\|\mathbf{e}\|^2} = 1 - \frac{\|(I - U_k U_k^T)E(I - U_k U_k^T)\|_F^2}{\|E\|_F^2}$$

where $U_k = [u_1, \ldots, u_k]$, and that

$$\lambda_{\min}^2(\boldsymbol{\Sigma}_k) = \min_{\|y_k\| = 1} \left\| \frac{u_k y_k^T + y_k u_k^T}{\sqrt{1 + \eta/\sigma_k^2}} \right\|_F^2 = \min_{\|y_k\| = 1} \frac{2\|u_k\|^2 \|y_k\|^2 + 2(u_k^T y_k)^2}{1 + \eta/\sigma_k^2} = \frac{2}{1 + \eta/\sigma_k^2}.$$

$\qquad \square$

## D.2    Proof of Basis Alignment (Lemma 13)

Before we prove this lemma, we make two observations that simplifies the proof. First, even though our goal is to prove the inequality (36) for all $k \geq r^*$, it actually suffices to consider the case $k = r^*$. This is because the numerator $\|Z^T(I - U_k U_k^T)Z\|_F$ decreases monotonically as $k$ increases. Indeed, for any $k \geq r^\star$, define $VV^T$ as below

$$I - U_k U_k^T = I - U_{r^\star} U_{r^\star}^T - VV^T = (I - U_{r^\star} U_{r^\star}^T)(I - VV^T) = (I - VV^T)(I - U_{r^\star} U_{r^\star}^T).$$

Then, we have

$$\|Z^T(I - U_kU_k^T)Z\|_F = \|(I - U_kU_k^T)ZZ^T(I - U_kU_k^T)\|_F$$
$$= \|(I - VV^T)(I - U_{r^\star}U_{r^\star}^T)ZZ^T(I - U_{r^\star}U_{r^\star}^T)(I - VV^T)\|_F$$
$$\leq \|(I - U_{r^\star}U_{r^\star}^T)ZZ^T(I - U_{r^\star}U_{r^\star}^T)\|_F.$$

Second, due to the rotational invariance of this problem, we can assume without loss of generality that $X, Z$ are of the form

$$X = \begin{bmatrix} X_1 & 0 \\ 0 & X_2 \end{bmatrix}, \; Z = \begin{bmatrix} Z_1 \\ Z_2 \end{bmatrix}. \tag{45}$$

where $X_1 \in \mathbb{R}^{k \times k}$, $Z_1 \in \mathbb{R}^{k \times r^\star}$ and $\sigma_{\min}(X_1) \geq \sigma_{\max}(X_2)$. (Concretely, we compute the singular value decomposition $X = USV^T$ with $U \in \mathbb{R}^{n \times n}$ and $V \in \mathbb{R}^{r \times r}$, and then set $X \leftarrow U^TXV$ and $Z \leftarrow U^TZ$.) We first need to show that as $XX^T$ approaches $ZZ^T$, the dominant directions of $X$ must align with $Z$ in a way as to make the $Z_2$ portion of $Z$ go to zero.

**Lemma 15.** *Suppose that $X, Z$ are in the form in (45), and $k \geq r^\star$. If $\|XX^T - ZZ^T\|_F \leq \rho\lambda_{\min}(Z^TZ)$ and $\rho^2 < 1/2$, then $\lambda_{\min}(Z_1^TZ_1) \geq \lambda_{\max}(Z_2^TZ_2)$.*

*Proof.* Denote $\gamma = \lambda_{\min}(Z_1^TZ_1)$ and $\beta = \lambda_{\max}(Z_2^TZ_2)$. We will assume $\gamma < \beta$ and prove that $\rho^2 \geq 1/2$, which contradicts our hypothesis. The claim is invariant to scaling of $X$ and $Z$, so we assume without loss of generality that $\lambda_{\min}(Z^TZ) = 1$. Our radius hypothesis then reads

$$\|XX^T - ZZ^T\|_F^2 = \left\| \begin{bmatrix} X_1X_1^T - Z_1Z_1^T & -Z_1Z_2^T \\ -Z_2Z_1^T & X_2X_2^T - Z_2Z_2^T \end{bmatrix} \right\|_F^2$$
$$= \|X_1X_1^T - Z_1Z_1^T\|_F^2 + 2\langle Z_1^TZ_1, Z_2^TZ_2\rangle + \|X_2X_2^T - Z_2Z_2^T\|_F^2 \leq \rho^2.$$

Now, we optimize over $X_1$ and $X_2$ to minimize the left-hand side. Recall by construction in (45) we restricted $\sigma_{\min}(X_1) \geq \sigma_{\max}(X_2)$. Accordingly, we consider

$$\min_{X_1, X_2} \left\{ \|X_1X_1^T - Z_1Z_1^T\|_F^2 + \|X_2X_2^T - Z_2Z_2^T\|_F^2 : \lambda_{\min}(X_1X_1^T) \geq \lambda_{\max}(X_2X_2^T) \right\}. \tag{46}$$

We relax $X_1X_1^T$ and $X_2X_2^T$ into positive semidefinite matrices

$$(46) \geq \min_{S_1 \succeq 0, S_2 \succeq 0}\{\|S_1 - Z_1Z_1^T\|_F^2 + \|S_2 - Z_2Z_2^T\|_F^2 : \lambda_{\min}(S_1) \geq \lambda_{\max}(S_2)\} \tag{47}$$

The equation above is invariant to a change of basis for both $S_1$ and $S_2$, so we change the basis of $S_1$ and $S_2$ into the eigenbases of $Z_1Z_1^T$ and $Z_2Z_2^T$ to yield

$$(47) = \min_{s_1 \geq 0, s_2 \geq 0}\{\|s_1 - \lambda(Z_1Z_1^T)\|^2 + \|s_2 - \lambda(Z_2Z_2^T)\|^2 : \min(s_1) \geq \max(s_2)\} \tag{48}$$

where $\lambda(Z_1Z_1^T) \geq 0$ and $\lambda(Z_2Z_2^T) \geq 0$ are the vector of eigenvalues. We lower-bound (48) by dropping all the terms in the sum of squares except the one associated with $\lambda_{\min}(Z_1^TZ_1)$ and $\lambda_{\max}(Z_2Z_2^T)$ to obtain

$$(48) \geq \min_{d_1, d_2 \in \mathbb{R}_+} \{[d_1 - \lambda_{\min}(Z_1^TZ_1)]^2 + [d_2 - \lambda_{\max}(Z_2Z_2^T)]^2 : d_1 \geq d_2\} \tag{49}$$

$$= \min_{d_1, d_2 \in \mathbb{R}_+} \{[d_1 - \gamma]^2 + [d_2 - \beta]^2 : d_1 \geq d_2\} = (\gamma - \beta)^2/2, \tag{50}$$

where we use the fact that $\gamma < \beta$ to argue that $d_1 = d_2$ at optimality. Now we have

$$\rho^2 \geq \|X_1X_1^T - Z_1Z_1^T\|_F^2 + \|X_2X_2^T - Z_2Z_2^T\|_F^2 + 2\langle Z_1^TZ_1, Z_2^TZ_2\rangle$$
$$\geq \|X_1X_1^T - Z_1Z_1^T\|_F^2 + \|X_2X_2^T - Z_2Z_2^T\|_F^2 + 2\lambda_{\min}(Z_1^TZ_1)\lambda_{\max}(Z_2^TZ_2)$$
$$\geq \min_{d_1, d_2 \in \mathbb{R}_+} \{[d_1 - \gamma]^2 + [d_2 - \beta]^2 : d_1 \geq d_2\} + 2\gamma\beta$$
$$\geq \frac{(\gamma - \beta)^2}{2} + 2\gamma\beta = \frac{1}{2}(\gamma + \beta)^2.$$

Finally, note that

$$\gamma + \beta = \lambda_{\min}(Z_1^TZ_1) + \lambda_{\max}(Z_2^TZ_2) \geq \lambda_{\min}(Z_1^TZ_1 + Z_2^TZ_2) = \lambda_{\min}(Z^TZ) = 1.$$

Therefore, we have $\rho^2 \geq 1/2$, a contradiction. This completes the proof. $\qquad\square$

Now we are ready to prove Lemma 13.

*Proof.* As before, assume with out loss of generality that $X, Z$ are of the form (45). From the proof of Lemma 15 we already know

$$\|XX^T - ZZ^T\|_F^2 = \|X_1 X_1^T - Z_1 Z_1^T\|_F^2 + 2\langle Z_1^T Z_1, Z_2^T Z_2\rangle + \|X_2 X_2^T - Z_2 Z_2^T\|_F^2.$$

Moreoever, we can compute

$$\|Z^T(I - U_k U_k^T)Z\|_F = \left\|\begin{bmatrix} Z_1 \\ Z_2 \end{bmatrix}^T \left(I - \begin{bmatrix} I_k & 0 \\ 0 & 0 \end{bmatrix}\right)\begin{bmatrix} Z_1 \\ Z_2 \end{bmatrix}\right\|_F = \|Z_2^T Z_2\|_F = \|Z_2 Z_2^T\|_F. \quad (51)$$

We will show that in the neighborhood $\|XX^T - ZZ^T\| \le \rho\lambda_{\min}(Z^T Z)$ that

$$\rho \le 1/\sqrt{2} \implies \sin\phi \overset{\text{def}}{=} \|(I - U_k U_k^T)Z\|_F/\sigma_k(Z) = \|Z_2\|_F/\sigma_{r^\star}(Z) \le \rho. \quad (52)$$

Then we obtain

$$\frac{\|Z_2 Z_2^T\|_F^2}{\|XX^T - ZZ^T\|^2} \overset{(a)}{\le} \frac{\|Z_2\|_F^4}{2\langle Z_1^T Z_1, Z_2^T Z_2\rangle} \overset{(b)}{\le} \frac{\|Z_2\|_F^4}{2\lambda_{\min}(Z_1^T Z_1)\|Z_2\|_F^2}$$

$$\overset{(c)}{\le} \frac{\|Z_2\|_F^2}{2[\lambda_{\min}(Z^T Z) - \|Z_2\|_F^2]} = \frac{\sin^2\phi}{2[1 - \sin^2\phi]} \quad (53)$$

$$\le \frac{1}{2}\frac{\rho^2}{1 - \rho^2}. \quad (54)$$

Step (a) bounds the numerator as $\|Z_2 Z_2^T\|_F \le \|Z_2\|_F^2$ and uses the fact that the denominator is greater than $2\langle Z_1^T Z_1, Z_2^T Z_2\rangle$. Step (b) follows from the inequality $\langle Z_1^T Z_1, Z_2^T Z_2\rangle \ge \lambda_{\min}(Z_1^T Z_1)\|Z_2 Z_2^T\|_F$. Finally, step (c) bounds the minimum eigenvalue of $Z_1^T Z_1$ by noting that

$$\lambda_{\min}(Z_1^T Z_1) = \lambda_{\min}(Z_1^T Z_1 + Z_2^T Z_2 - Z_2^T Z_2)$$
$$\ge \lambda_{\min}(Z_1^T Z_1 + Z_2^T Z_2) - \lambda_{\max}(Z_2^T Z_2)$$
$$\ge \lambda_{\min}(Z^T Z) - \|Z_2\|_F^2, \quad (55)$$

where the last line bounds the operator norm of $Z_2$ with the Frobenius norm.

To prove (52), we know from Lemma 15 that $\rho \le 1/\sqrt{2}$ implies that $\lambda_{\min}(Z_1^T Z_1) \ge \lambda_{\max}(Z_2^T Z_2)$. This implies $\lambda_{\min}(Z_1^T Z_1) \ge \frac{1}{2}\lambda_{\min}(Z^T Z)$, since

$$2\lambda_{\min}(Z_1^T Z_1) \ge \lambda_{\min}(Z_1^T Z_1) + \lambda_{\max}(Z_2^T Z_2) \ge \lambda_{\min}(Z^T Z)$$

This implies the following

$$\|XX^T - ZZ^T\|_F^2 = \|X_1 X_1^T - Z_1 Z_1^T\|_F^2 + 2\langle Z_1^T Z_1, Z_2^T Z_2\rangle + \|X_2 X_2^T - Z_2 Z_2^T\|_F^2$$
$$\ge 2\langle Z_1^T Z_1, Z_2^T Z_2\rangle \ge 2\lambda_{\min}(Z_1^T Z_1)\|Z\|_F^2 \ge \lambda_{\min}(Z^T Z)\|Z\|_F^2$$

and we have therefore

$$\rho^2\lambda_{\min}^2(Z^T Z) \ge \|XX^T - ZZ^T\|_F^2 \ge \lambda_{\min}(Z^T Z)\|Z\|_F^2 \ge \lambda_{\min}(Z^T Z)\|Z_2\|_F^2$$

which this proves $\sin^2\phi = \|Z_2\|_F^2/\lambda_{\min}(Z^T Z) \le \rho^2$ as desired. $\qquad\square$

# E  Preliminaries for the Noisy Case

## E.1  Notations

In the following sections, we extend our proofs to the noisy setting. As before, we denote by $M^\star = ZZ^T \in \mathbb{R}^{n\times n}$ our ground truth. Our measurements are of the form $y = \mathcal{A}(ZZ^T) + \epsilon \in \mathbb{R}^m$. We make the standard assumption that the noise vector $\epsilon \in \mathbb{R}^m$ has sub-Gaussian entries with zero mean and variance $\sigma^2 = \frac{1}{m}\sum_{i=1}^m \mathbb{E}[\epsilon_i^2]$.

In this case, the objective function can be written as

$$f(X) = \frac{1}{m}\|\mathcal{A}(XX^T) - y\|^2 = f_c(X) + \frac{1}{m}\|\epsilon\|^2 - \frac{2}{m}\langle\mathcal{A}(XX^T - M^\star), \epsilon\rangle,$$

where $f_c(X) = \frac{1}{m}\|\mathcal{A}(XX^T - M^\star)\|^2$ is the objective function with clean measurements that are not corrupted with noise. Note that compared to the noiseless case, we have rescaled our objective by a factor of $1/m$ to emphasize the number of measurements $m$.

Moreover, we say that an event $\mathcal{E}$ happens with overwhelming or high probability, if its probability of occurrence is at least $1 - cn^{-c'}$, for some $0 < c, c' < \infty$. Moreover, to streamline the presentation, we omit the statement "with high or overwhelming probabily" if it is implied by the context.

We make a few simplifications on notations. As before, we will use $\alpha$ to denote the step-size and $D$ to denote the local search direction. We will use lower case letters $x$ and $d$ to refer to $\text{vec}(X)$ and $\text{vec}(D)$ respectively.

Similarly, we will write $f(x) \in \mathbb{R}^{nr}$ and $\nabla f(x) \in R^{nr}$ as the vectorized versions of $f(X)$ and its gradient. This notation is also used for $f_c(X)$. As before, we define $P = X^T X + \eta I_r$ and $\mathbf{P} = (X^T X + \eta I_r) \otimes I_n$. For the vectorized version of the gradient, we simply define its $P$-norm (and $P^*$-norm) to be the same as the matrix version, that is,

$$\|\nabla f(x)\|_P = \|\nabla f(X)\|_P, \qquad \|\nabla f(x)\|_{P^*} = \|\nabla f(X)\|_{P^*}.$$

We drop the iteration index $k$ from our subsequent analysis, and refer to $x_{k+1}$ and $x_k$ as $\tilde{x}$ and $x$, respectively. Thus, with noisy measurements, the iterations of PrecGD take the form

$$X_{k+1} = X_k - \alpha\nabla f(X_k)(X_k^T X_k)^{-1}.$$

The *vectorized* version of the gradient update above can be written as $\tilde{x} = x - \alpha d$, where

$$d = \text{vec}(\nabla f(X)P^{-1}) = \text{vec}\left(f_c(X) + \frac{1}{m}\|\epsilon\|^2 - \frac{2}{m}\langle\mathcal{A}(XX^T - M^\star), \epsilon\rangle\right)$$
$$= \mathbf{P}^{-1}\nabla f_c(x) - \frac{2}{m}\mathbf{P}^{-1}\left(I_r \otimes \sum_{i=1}^{m}\epsilon_i A_i\right)x. \tag{56}$$

Inspired by the variational representation of the Frobenius norm, for any matrix $H \in \mathbb{R}^{n \times n}$ we define its *restricted Frobenius norm* as

$$\|H\|_{F,r} = \arg\max_{Y \in S_n^+, \text{rank}(Y) \leq r}\langle H, Y\rangle, \tag{57}$$

where $S_n^+$ is the set of $n \times n$ positive semidefinite matrices. It is easy to verify that $\|H\|_F = \|H\|_{F,n}$ and $\|H\|_{F,r} = \sqrt{\sum_{i=1}^{r}\sigma_i(H)^2}$.

For any two real numbers $a, b \in R$, we say that $a \asymp b$ if there exists some constant $C_1, C_2$ such that $C_1 b \leq a \leq C_2 b$. Through out the section we will use one symbol $C$ to denote constants that might differ.

Finally, we also recall that $\mu_P$, which is used repeatedly in this section, is the constant defined in (33).

### E.2   Auxiliary Lemmas

Now we present a few auxiliary lemmas that we will use for the proof of the noisy case. At the core of our subsequent proofs is the following standard concentration bound.

**Lemma 16.** *Suppose that the number of measurements satisfies $m \gtrsim \sigma n \log n$. Then, with high probability, we have*

$$\frac{1}{m}\left\|\sum_{i=1}^{m}A_i\epsilon_i\right\|_2 \lesssim \sqrt{\frac{\sigma^2 n \log n}{m}},$$

*where $\|\cdot\|_2$ denotes the operator norm of a matrix.*

Lemma 16 will be used extensively in the proofs of Proposition 6, and Theorems 7 and 8.

Our strategy for establishing linear convergence is similar to that of the noiseless case. Essentially, our goal is to show that with an appropriate step-size, there is sufficient decrement in the objective value in terms of $\|\nabla f_c(X)\|_{P^*}$. Then applying Theorem 4 will result in the desired convergence rate.

In the noiseless case, we proved a Lipschitz-like inequality (Lemma 2) and bounded the Lipschitz constant above in a neighborhood around the ground truth. Similar results hold in the noisy case. However, because of the noise, it will be easier to directly work with the quartic polynomial $f_c(X - \alpha D)$ instead. In particular, we have the following lemma that characterizes how much progress we make by taking a step in the direction $D$.

**Lemma 17.** *For any descent direction $D \in \mathbb{R}^{n \times r}$ and step-size $\alpha > 0$ we have*

$$f_c(X - \alpha D) \le f_c(X) - \alpha \nabla f_c(X)^T D + \frac{\alpha^2}{2} D^T \nabla^2 f_c(X) D \tag{58}$$

$$+ \frac{(1+\delta)\alpha^3}{m} \|D\|_F^2 \left( 2\|DX^T + XD^T\|_F + \alpha \|D\|_F^2 \right). \tag{59}$$

*Proof.* Directly expanding the quadratic $f_c(X - \alpha D)$, we get

$$f_c(X - \alpha D) = \frac{1}{m} \|\mathcal{A}((X - \alpha D)(X - \alpha D)^T - M^\star)\|^2$$

$$= \frac{1}{m} \|\mathcal{A}(XX^T - M^\star)\|^2 - \frac{2\alpha}{m} \langle \mathcal{A}(XX^T - M^\star), \mathcal{A}(XD^T + DX^T) \rangle$$

$$+ \frac{\alpha^2}{m} \left[ 2\langle \mathcal{A}(XX^T - M^\star), \mathcal{A}(DD^T) \rangle + \|\mathcal{A}(XD^T + DX^T)\|^2 \right]$$

$$- \frac{2\alpha^3}{m} \langle \mathcal{A}(XD^T + DX^T), \mathcal{A}(DD^T) \rangle + \frac{\alpha^4}{m} \|\mathcal{A}(DD^T)\|^2.$$

We bound the third- and fourth- order terms

$$|\langle \mathcal{A}(XD^T + DX^T), \mathcal{A}(DD^T) \rangle| \overset{(a)}{\le} \|\mathcal{A}(XD^T + DX^T)\| \|\mathcal{A}(DD^T) \rangle\|$$

$$\overset{(b)}{\le} (1+\delta) \|XD^T + DX^T\|_F \|DD^T\|_F$$

$$\overset{(c)}{\le} (1+\delta) \|XD^T + DX^T\|_F \|D\|_F^2$$

and

$$\|\mathcal{A}(DD^T)\|^2 \overset{(b)}{\le} (1+\delta) \|DD^T\|_F^2 \overset{(c)}{\le} (1+\delta) \|D\|_F^4,$$

Step (a) uses the Cauchy–Schwarz inequality; Step (b) applies $(\delta, 2r)$-RIP; Step (c) bounds $\|DD^T\|_F \le \|D\|_F^2$. Summing up these inequalities we get the desired result. $\qquad\square$

It turns out that in our proofs it will be easier to work with the *vectorized* version of (59), which we can write as

$$f_c(x - \alpha d) \le f_c(x) - \alpha \nabla f_c(x)^T d + \frac{\alpha^2}{2} d^T \nabla^2 f_c(x) d + \frac{(1+\delta)\alpha^3}{m} \|d\|^2 \left( 2\|\mathbf{J}_X d\| + \alpha \|d\|^2 \right), \tag{60}$$

where we recall that $J_X : \mathbb{R}^{nr} \to \mathbb{R}^{n^2}$ is the linear operator that satisfies $J_X d = \mathrm{vec}(XD^T + DX^T)$.

Now we proceed to bound the higher-order terms in the Taylor-like expansion above.

**Lemma 18** (Second-order term). *We have*

$$\sigma_{\max}(\mathbf{P}^{-1/2} \nabla^2 f_c(x) \mathbf{P}^{-1/2}) \le \frac{2(1+\delta)}{m} \left( \frac{8\sigma_r^2(X) + \|XX^T - ZZ^T\|_F}{\sigma_r^2(X) + \eta} \right).$$

*Proof.* For any $v \in \mathbb{R}^{nr}$ where $v = \mathrm{vec}(V)$, we have

$$m \cdot v^T \nabla^2 f_c(x) v = 4\langle \mathcal{A}(XX^T - ZZ^T), \mathcal{A}(VV^T) + 2\|\mathcal{A}(XV^T + VX^T)\|^2$$

$$\le 4\|\mathcal{A}(XX^T - ZZ^T)\| \|\mathcal{A}(VV^T)\| + 2\|\mathcal{A}(XV^T + VX^T)\|^2$$

$$\le 2(1+\delta) \left( \|XX^T - ZZ^T\|_F \|VV^T\|_F + 2\|XV^T + VX^T\|_F^2 \right)$$

Now, let $v = \mathbf{P}^{-1/2}u$ for $u = \text{vec}(U)$. Then, $V = UP^{-1/2}$ and

$$\|VV^T\|_F = \|UP^{-1}U^T\|_F \leq \sigma_{\max}(P^{-1})\|U\|_F^2 = \frac{\|U\|_F^2}{\sigma_r^2(X) + \eta}.$$

Also, $\|XV^T + VX^T\|_F \leq 2\|XV^T\|_F$ and

$$\|XV^T\| = \|XP^{-1/2}U^T\| \leq \sigma_{\max}(XP^{-1/2})\|U\|_F = \left(\frac{\sigma_r^2(X)}{\sigma_r^2(X) + \eta}\right)^{1/2}\|U\|_F.$$

Since $\|u\| = \|U\|_F$, it follows that

$$u^T\mathbf{P}^{-1/2}\nabla^2 f_c(x)\mathbf{P}^{-1/2}u \leq \frac{2(1+\delta)}{m}\left(\frac{8\sigma_r^2(X) + \|XX^T - ZZ^T\|}{\sigma_r^2(X) + \eta}\right)\|u\|^2,$$

which gives the desired bound on the largest singular value. $\qquad\square$

The following lemma gives a bound on the third- and fourth-order terms in (60).

**Lemma 19.** *Set* $d = \mathbf{P}^{-1}\nabla f_c(x)$, *then we have* $\|\mathbf{J}d\|^2 \leq 8m^2\|\nabla f_c(x)\|_{P^*}^2$ *and* $\|d\|^2 \leq \|\nabla f_c(x)\|_{P^*}^2/\eta$.

*Proof.* We have

$$\begin{aligned}
\|\mathbf{J}_X d\|^2 &= \|\mathcal{A}(XD^T + DX^T)\|^2 \leq (1+\delta)\|XD^T + DX^T\|^2 \\
&= (1+\delta)\|\mathbf{J}_X d\|^2 = m^2(1+\delta)\|\mathbf{J}\mathbf{P}^{-1}\nabla f_c(x)\|^2 \\
&\leq m^2(1+\delta)\sigma_{\max}^2(\mathbf{J}\mathbf{P}^{-1/2})\|\mathbf{P}^{-1/2}\nabla f_c(x)\|^2 \\
&= 4m^2(1+\delta)\frac{\sigma_r^2}{\sigma_r^2 + \eta}\|\nabla f_c(x)\|_{P^*}^2 \leq 8m^2\|\nabla f_c(x)\|_{P^*}^2
\end{aligned}$$

and

$$\begin{aligned}
\|d\|^2 = \|\mathbf{P}^{-1}\nabla f_c(x)\|^2 &\leq \sigma_{\max}(\mathbf{P}^{-1})\|\mathbf{P}^{-1/2}\nabla f_c(x)\|^2 \\
&= \frac{1}{\sigma_r^2 + \eta}\|\nabla f(x)\|_{P^*}^2 \leq \|\nabla f(x)\|_{P^*}^2/\eta.
\end{aligned}$$

$\square$

# F  Proof of Noisy Case with Optimal Damping Parameter

Now we are ready to prove Theorem 7, which we restate below for convenience.

**Theorem 20** (Noisy measurements with optimal $\eta$). *Suppose that the noise vector* $\epsilon \in \mathbb{R}^m$ *has sub-Gaussian entries with zero mean and variance* $\sigma^2 = \frac{1}{m}\sum_{i=1}^m \mathbb{E}[\epsilon_i^2]$. *Moreover, suppose that* $\eta_k = \frac{1}{\sqrt{m}}\|\mathcal{A}(X_k X_k^T - M^*)\|$, *for* $k = 0, 1, \ldots, K$, *and that the initial point* $X_0$ *satisfies* $\|\mathcal{A}(X_0 X_0^T - M^*)\|^2 < \rho^2(1-\delta)\lambda_{r^*}(M^\star)^2$. *Consider* $k^* = \arg\min_k \eta_k$, *and suppose that* $\alpha \leq 1/L$, *where* $L > 0$ *is a constant that only depends on* $\delta$. *Then, with high probability, we have*

$$\|X_{k^*}X_{k^*}^T - M^*\|_F^2 \lesssim \max\left\{\frac{1+\delta}{1-\delta}\left(1 - \alpha\frac{\mu_P}{2}\right)^K \|X_0 X_0^T - M^*\|_F^2, \mathcal{E}_{stat}\right\}, \quad (61)$$

*where* $\mathcal{E}_{stat} := \frac{\sigma^2 nr \log n}{\mu_P(1-\delta)m}$.

*Proof.* **Step I. Using Lemma 17 to establish sufficient decrement.**

First, we write out the vectorized version of Lemma 60:

$$f_c(x - \alpha d) \leq f_c(x) - \alpha\nabla f_c(x)^T d + \frac{\alpha^2}{2}d^T\nabla^2 f_c(x)d + \frac{(1+\delta)\alpha^3}{m}\|d\|^2\left(2\|\mathbf{J}_X d\| + \alpha\|d\|^2\right). \quad (62)$$

To simplify notation, we define the error term $\mathbb{E}(x) = \frac{2}{m}\left(I_r \otimes \sum_{i=1}^m \epsilon_i A_i\right)x$, so that the search direction (56) can be rewritten as $d = \mathbf{P}^{-1}(\nabla f_c(x) - \mathbb{E}(x))$.

Now plugging this $d$ into (62) yields

$$f_c(x - \alpha d) \leq f_c(x) - \alpha \|\nabla f_c(x)\|_{P*}^2 + T_1 + T_2 + T_3$$

where

$$
\begin{aligned}
T_1 =& \alpha \nabla f_c(x)^T \mathbf{P}^{-1} \mathbb{E}(x) \\
T_2 =& \frac{\alpha^2}{2} \left( \nabla f_c(x)^T \mathbf{P}^{-1} \nabla^2 f_c(x) \mathbf{P}^{-1} \nabla f_c(x) + \mathbb{E}(x)^T \mathbf{P}^{-1} \nabla^2 f_c(x) \mathbf{P}^{-1} \mathbb{E}(x) \right. \\
& \left. - 2 \nabla f_c(x)^T \mathbf{P}^{-1} \nabla^2 f_c(x) \mathbf{P}^{-1} \mathbb{E}(x) \right) \\
T_3 =& (1 + \delta) \alpha^3 \left( \|\mathbf{P}^{-1} \nabla f_c(x) - \mathbf{P}^{-1} \mathbb{E}(x)\|^2 \right) \left( 2 \|\mathbf{J} \mathbf{P}^{-1} \nabla f_c(x)\| + 2 \|\mathbf{J} \mathbf{P}^{-1} \mathbb{E}(x)\| \right. \\
& \left. + \alpha \|\mathbf{P}^{-1} \nabla f_c(x) - \mathbf{P}^{-1} \mathbb{E}(x)\|^2 \right).
\end{aligned}
$$

## II. Bounding $T_1, T_2$ and $T_3$.

We control each term in the above expression individually. First, we have

$$T_1 = \alpha \nabla f_c(x)^T \mathbf{P}^{-1} \mathbb{E}(x) \leq \alpha \|\mathbf{P}^{-1} \nabla f_c(x)\|_P \|\mathbb{E}(x)\|_{P*} = \alpha \|\nabla f_c(x)\|_{P*} \|\mathbb{E}(x)\|_{P*}.$$

To bound $T_2$, first we note that for any vectors $x, y \in \mathbb{R}^n$ and any positive semidefinite matrix $P \in S_+^n$, we always have $(x + y)^T P (x + y) \leq 2(x^T P x + y^T P y)$. Therefore we can bound

$$T_2 \leq \alpha^2 \left( \nabla f_c(x)^T \mathbf{P}^{-1} \nabla^2 f_c(x) \mathbf{P}^{-1} \nabla f_c(x) + \mathbb{E}(x)^T \mathbf{P}^{-1} \nabla^2 f_c(x) \mathbf{P}^{-1} \mathbb{E}(x) \right).$$

Next, we apply Lemma 18 to arrive at

$$\frac{1}{2} \sigma_{\max}(\mathbf{P}^{-1/2} \nabla^2 f_c(x) \mathbf{P}^{-1/2}) \leq \frac{1 + \delta}{m} \left( \frac{8 \sigma_r^2(X) + \|XX^T - M^\star\|}{\sigma_r^2(X) + \eta} \right) \stackrel{def}{\leq} L_\delta,$$

where $L_\delta$ is a constant that only depends on $\delta$ and $m$. Note that the last inequality follows from the fact that $\eta = O(\|XX^T - M^\star\|)$.

Now based on the above inequality, we have

$$
\begin{aligned}
\alpha^2 \left( \nabla f_c(x)^T \mathbf{P}^{-1} \nabla^2 f_c(x) \mathbf{P}^{-1} \nabla f_c(x) \right) &\leq 2\alpha^2 L_\delta \|\nabla f_c(x)\|_{P*}^2 \\
\alpha^2 \left( \mathbb{E}(x)^T \mathbf{P}^{-1} \nabla^2 f_c(x) \mathbf{P}^{-1} \mathbb{E}(x) \right) &\leq 2\alpha^2 L_\delta \|\mathbb{E}(x)\|_{P*}^2,
\end{aligned}
$$

which implies

$$T_2 \leq 2\alpha^2 L_\delta \|\nabla f_c(x)\|_{P*}^2 + 2\alpha^2 L_\delta \|\mathbb{E}(x)\|_{P*}^2$$

Finally, to bound $T_3$, we first write

$$\|\mathbf{P}^{-1} \nabla f_c(x) - \mathbf{P}^{-1} \mathbb{E}(x)\|^2 \leq 2 \|\mathbf{P}^{-1} \nabla f_c(x)\|^2 + 2 \|\mathbf{P}^{-1} \mathbb{E}(x)\|^2.$$

Moreover, invoking Lemma 19 leads to the following inequalities

$$
\begin{aligned}
\|\mathbf{P}^{-1} \nabla f_c(x)\|^2 &\leq \frac{\|\nabla f_c(x)\|_{P*}^2}{\eta}, & \|\mathbf{P}^{-1} \mathbb{E}(x)\|^2 &\leq \frac{\|\mathbb{E}(x)\|_{P*}^2}{\eta}. \\
\|\mathbf{J} \mathbf{P}^{-1/2} \nabla f_c(x)\| &\leq 2\sqrt{2} \|\nabla f_c(x)\|_{P*}, & \|\mathbf{J} \mathbf{P}^{-1/2} \mathbb{E}(x)\| &\leq 2\sqrt{2} \|\mathbb{E}(x)\|_{P*}.
\end{aligned}
$$

Combining the above inequalities with the definition of $T_3$ leads to:

$$
\begin{aligned}
T_3 \leq & \frac{4(1 + \delta) \alpha^3}{\eta} \left( \|\nabla f_c(x)\|_{P*}^2 + \|\mathbb{E}(x)\|_{P*}^2 \right) \\
& \times \left( 2\sqrt{2} \|\nabla f_c(x)\|_{P*} + 2\sqrt{2} \|\nabla \mathbb{E}(x)\|_{P*} + \frac{\alpha}{\eta} \|\nabla f_c(x)\|_{P*}^2 + \frac{\alpha}{\eta} \|\mathbb{E}(x)\|_{P*}^2 \right).
\end{aligned}
$$

## III. Bounding the Error Term

Next, we provide an upper bound on $\|\mathbb{E}(x)\|_{P^*}$. The following chain of inequalities hold with high probability:

$$
\begin{aligned}
\|\mathbb{E}(x)\|_{P^*}^2 = \mathbb{E}(x)^T \mathbf{P}^{-1} \mathbb{E}(x) &= \left\| \left( \frac{2}{m} \sum_{i=1}^m \epsilon_i A_i \right) X (X^T X + \eta I)^{-1/2} \right\|_F^2 \\
&\leq \left\| \left( \frac{2}{m} \sum_{i=1}^m \epsilon_i A_i \right) \right\|_2^2 \left\| X (X^T X + \eta I)^{-1/2} \right\|_F^2 \\
&\overset{(a)}{\leq} C \frac{\sigma^2 n \log n}{m} \left( \sum_{i=1}^r \frac{\sigma_i^2(X)}{\sigma_i(X)^2 + \eta} \right) \\
&\leq C \frac{\sigma^2 r n \log n}{m},
\end{aligned}
$$

where $C$ is an absolute constant and (a) follows from Lemma 16.

**IV. Bounding all the terms using $\|\nabla f_c(x)\|_{P*}$**

Combining the upper bound on $\|\mathbb{E}(X)\|_{P*}$ with the previous bounds for $T_1, T_2, T_3$ and denoting $\Delta = \|\nabla f_c(x)\|_{P*}$, we have

$$
T_1 \leq \alpha \Delta \sqrt{\frac{C \sigma^2 r n \log n}{m}},
$$

$$
T_2 \leq 2\alpha^2 L_\delta \Delta^2 + 2\alpha^2 L_\delta \frac{\sigma^2 r n \log n}{m}
$$

$$
T_3 \leq \frac{4(1+\delta)\alpha^3}{\eta} \left( \Delta^2 + \frac{C \sigma^2 r n \log n}{m} \right) \left( \frac{\alpha \Delta^2}{\eta} + \frac{\alpha C \sigma^2 r n \log n}{\eta m} + 2\sqrt{2}\Delta + 2\sqrt{2} \sqrt{\frac{C \sigma^2 r n \log n}{m}} \right)
$$

Now, combining the upper bounds for $T_1, T_2$ and $T_3$ with (62) yields

$$
\begin{aligned}
f_c(x - \alpha d) \leq f_c(x) &- \alpha \Delta^2 + \alpha \Delta \sqrt{\frac{C \sigma^2 r n \log n}{m}} + 2\alpha^2 L_\delta \Delta^2 + 2C\alpha^2 L_\delta \frac{\sigma^2 r n \log n}{m} \\
&+ \frac{4(1+\delta)\alpha^3}{\eta} \left( \Delta^2 + \frac{C \sigma^2 r n \log n}{m} \right) \left( \frac{\alpha \Delta^2}{\eta} + \frac{\alpha C \sigma^2 r n \log n}{\eta m} + 2\sqrt{2}\Delta + 2\sqrt{2} \sqrt{\frac{C \sigma^2 r n \log n}{m}} \right).
\end{aligned}
\tag{63}
$$

The above inequality holds with high probability for every iteration of PrecGD.

**V. Two cases**

Now, we consider two cases. First, suppose that $\eta \leq 2\sqrt{\frac{C\sigma^2 nr \log n}{\mu_P m}}$. This implies that $\min_k \eta_k \leq 2\sqrt{\frac{C\sigma^2 nr \log n}{\mu_P m}}$, and hence,

$$
\|X_{k^*} X_{k^*}^T - M^\star\|_F^2 \lesssim \frac{1}{1-\delta} \frac{1}{m} \|\mathcal{A}(X_{k^*} X_{k^*}^T - M^\star)\|^2 \lesssim \mathcal{E}_{stat}
$$

which completes the proof.

Otherwise, suppose that $\eta > 2\sqrt{\frac{C\sigma^2 nr \log n}{\mu_P m}}$. Due to Theorem 4, we have $\Delta \geq 2\sqrt{\frac{C\sigma^2 rn \log n}{m}}$, which leads to the following inequalities:

$$
-\alpha \Delta^2 + \alpha \Delta \sqrt{\frac{C \sigma^2 r n \log n}{m}} \leq -\frac{\alpha}{2} \Delta^2, \qquad 2\alpha^2 L_\delta \Delta^2 + 2C\alpha^2 L_\delta \frac{\sigma^2 r n \log n}{m} \leq \frac{5}{2}\alpha^2 L_\delta \Delta^2.
$$

Similarly, we have

$$
\Delta^2 + \frac{C \sigma^2 r n \log n}{m} \leq \frac{5}{4}\Delta^2, \quad 2\sqrt{2}\Delta + 2\sqrt{2} \sqrt{\frac{C \sigma^2 r n \log n}{m}} \leq 3\sqrt{2}\Delta,
$$

and

$$\frac{\alpha\Delta^2}{\eta} + \frac{\alpha}{\eta}\frac{C\sigma^2 rn\log n}{m} \leq \frac{5}{4}\frac{\alpha\Delta^2}{\eta}.$$

Combined with (63), we have

$$f_c(x - \alpha d) \leq f_c(x) - \frac{\alpha}{2}\Delta^2 + \frac{5}{2}\alpha^2 L_\delta \Delta^2 + \frac{4(1+\delta)\alpha^3}{\eta}\left(\frac{5}{4}\Delta^2\right)\left(3\sqrt{2}\Delta + \frac{5}{4}\frac{\alpha\Delta^2}{\eta}\right)$$

$$\leq f_c(x) - \frac{\alpha}{2}\Delta^2\left(1 - \frac{5}{2}L_\delta\alpha - 60\sqrt{2}\frac{\alpha^2\Delta}{\eta} - 25\alpha^3\left(\frac{\Delta}{\eta}\right)^2\right).$$

Similar to the noiseless case, we can bound the ratio $\frac{\Delta}{\eta}$ as

$$\frac{\Delta}{\eta} = \frac{\|\nabla f_c(x)\|_{P*}}{\eta} \leq \frac{(1+\delta)\sigma_{\max}(\mathbf{JP}^{-1/2})\|\mathbf{e}\|}{\|\mathbf{e}\|} = (1+\delta)\frac{\sigma_{\max}^2(X)}{\sigma_{\max}^2(X) + \eta} \leq 1 + \delta,$$

which in turn leads to

$$f_c(x - \alpha d) \leq f_c(x) - \frac{\alpha}{2}\Delta^2\left(1 - \frac{5}{2}L_\delta\alpha - 60\sqrt{2}\alpha^2(1+\delta) - 25\alpha^3(1+\delta)^2\right).$$

Now, assuming that the step-size satisfies $\alpha \leq \min\left\{\frac{L_\delta}{60\sqrt{2}(1+\delta) + 25(1+\delta)^2}, \frac{1}{7L_\delta}\right\}$. Since $L_\delta$ is a constant, we can simply write the condition above as $\alpha \leq 1/L$ where $L = \max\left\{\frac{60\sqrt{2}(1+\delta)+25(1+\delta)^2}{L_\delta}, 7L_\delta\right\}$. Now note that

$$\frac{5}{2}L_\delta + 60\sqrt{2}(1+\delta)\alpha + 25(1+\delta)^2\alpha^2 \leq \frac{7}{2}L_\delta$$

$$\implies 1 - \frac{5}{2}L_\delta\alpha - 60\sqrt{2}(1+\delta)\alpha^2 - 25(1+\delta)^2\alpha^3 \geq 1 - \frac{7}{2}L_\delta\alpha \geq \frac{1}{2}.$$

This implies that

$$f_c(x - \alpha d) \leq f_c(x) - \frac{t\Delta^2}{4} \leq \left(1 - \frac{\alpha\mu_P}{4}\right)f_c(x),$$

where in the last inequality, we used $\Delta^2 \geq \mu_P f_c(x)$, which is just the PL-inequality in Theorem 4. Finally, since $f_c(x)$ satisfies the RIP condition, combining the two cases above we get

$$\|X_{k^*}X_{k^*}^T - M^\star\|_F^2 \lesssim \max\left\{\frac{1+\delta}{1-\delta}\left(1 - \alpha\frac{\mu_P}{2}\right)^k\|X_0X_0^T - M^*\|_F^2, \mathcal{E}_{stat}\right\}, \qquad (64)$$

as desired. $\qquad\square$

# G  Proof of Noisy Case with Variance Proxy (Theorem 8)

In this section we prove Theorem 8, which we restate below for convenience. The only difference between this theorem and Theorem 7 is that we de not assume that we have access to the optimal choice of $\eta$. Instead, we only assume that we have some proxy $\hat{\sigma}^2$ of the true variance of the noise. For convenience we restate our result below.

**Theorem 21** (Noisy measurements with variance proxy). *Suppose that the noise vector $\epsilon \in \mathbb{R}^m$ has sub-Gaussian entries with zero mean and variance $\sigma^2 = \frac{1}{m}\sum_{i=1}^m \mathbb{E}[\epsilon_i^2]$. Moreover, suppose that $\eta_k = \sqrt{|f(X_k) - \hat{\sigma}^2|}$ for $k = 0, 1, \ldots, K$, where $\hat{\sigma}^2$ is an approximation of $\sigma^2$, and that the initial point $X_0$ satisfies $\|\mathcal{A}(X_0X_0^T - M^*)\|_F^2 < \rho^2(1-\delta)\lambda_{r^*}(M^\star)^2$. Consider $k^* = \arg\min_k \eta_k$, and suppose that $\alpha \leq 1/L$, where $L > 0$ is a constant that only depends on $\delta$. Then, with high probability, we have*

$$\|X_{k^*}X_{k^*}^T - M^*\|_F^2 \lesssim \max\left\{\frac{1+\delta}{1-\delta}\left(1 - \alpha\frac{\mu_P}{2}\right)^K\|X_0X_0^T - M^*\|_F^2, \mathcal{E}_{stat}, \mathcal{E}_{dev}, \mathcal{E}_{var}\right\}, \quad (65)$$

*where*

$$\mathcal{E}_{stat} := \frac{\sigma^2 nr\log n}{\mu_P(1-\delta)m}, \quad \mathcal{E}_{dev} := \frac{\sigma^2}{1-\delta}\sqrt{\frac{\log n}{m}}, \quad \mathcal{E}_{var} := |\sigma^2 - \hat{\sigma}^2|^2. \qquad (66)$$

The proof of Theorem 8 is similar to that of Theorem 7, with a key difference that $\eta_k = \frac{1}{\sqrt{m}}\|\mathcal{A}(X_k X_k^T - M^\star)\|$ is replaced with $\eta_k = \sqrt{|f(x_k) - \hat{\sigma}^2|}$. Our next lemma shows that this alternative choice of damping parameter remains close to $\frac{1}{\sqrt{m}}\|\mathcal{A}(X_k X_k^T - M^\star)\|$, provided that the error exceeds a certain threshold.

**Lemma 22.** *Set* $\eta = \sqrt{|f(x) - \hat{\sigma}^2|}$. *Then, with high probability, we have*

$$\sqrt{\frac{1/4 - \delta}{1 + \delta}} \frac{1}{\sqrt{m}} \left\| \mathcal{A}(XX^T - M^\star) \right\| \leq \eta \leq \sqrt{\frac{7/4 + \delta}{1 - \delta}} \frac{1}{\sqrt{m}} \left\| \mathcal{A}(XX^T - M^\star) \right\|$$

*provided that*

$$\|XX^T - M^\star\|_F^2 \gtrsim \max\left\{ \frac{\sigma^2 rn \log n}{m}, \sqrt{\frac{\sigma^2 \log n}{m}}, |\sigma^2 - \hat{\sigma}^2| \right\}.$$

*Proof.* One can write

$$f(x) = \frac{1}{m}\|y - \mathcal{A}(XX^T)\|^2 = \frac{1}{m}\|\mathcal{A}(M^\star - XX^T) + \epsilon\|^2$$
$$= \frac{1}{m}\|\mathcal{A}(M^\star - XX^T)\|^2 + \frac{1}{m}\|\epsilon\|^2 + \frac{2}{m}\left\langle \mathcal{A}(M^\star - XX^T), \epsilon \right\rangle.$$

Due to the definition of the restricted Frobenius norm (57), we have

$$|\left\langle \mathcal{A}(M^\star - XX^T), \epsilon \right\rangle| \leq \|M^\star - XX^T\|_F \left\| \frac{1}{m}\sum_{i=1}^m A_i \epsilon_i \right\|_{F,2r}.$$

Therefore, we have

$$\left| \frac{1}{m}\|\mathcal{A}(M^\star - XX^T)\|^2 + \frac{1}{m}\|\epsilon\|^2 - \hat{\sigma}^2 - 2\|M^\star - XX^T\|_F \left\| \frac{1}{m}\sum_{i=1}^m A_i \epsilon_i \right\|_{F,2r} \right| \leq \eta^2 \quad (67)$$

$$\left| \frac{1}{m}\|\mathcal{A}(M^\star - XX^T)\|^2 + \frac{1}{m}\|\epsilon\|^2 - \hat{\sigma}^2 + 2\|M^\star - XX^T\|_F \left\| \frac{1}{m}\sum_{i=1}^m A_i \epsilon_i \right\|_{F,2r} \right| \geq \eta^2. \quad (68)$$

Since the error $\epsilon_i$ is sub-Gaussian with parameter $\sigma$, the random variable $\epsilon_i^2$ is sub-exponential with parameter $16\sigma$. Therefore,

$$\mathbb{P}\left( \left| \frac{1}{m}\|\epsilon\|^2 - \sigma^2 \right| \geq t \right) \leq 2\exp\left( -\frac{Cmt^2}{\sigma^2} \right).$$

Now, upon setting $t = \sqrt{\frac{\sigma^2 \log n}{m}}$, we have

$$\left| \frac{1}{m}\|\epsilon\|^2 - \sigma^2 \right| \leq \sqrt{\frac{\sigma^2 \log n}{m}},$$

Moreover, we have

$$\left\| \frac{1}{m}\sum_{i=1}^m A_i \epsilon_i \right\|_{F,2r} \leq \sqrt{2r} \left\| \frac{1}{m}\sum_{i=1}^m A_i \epsilon_i \right\|_2 \lesssim \sqrt{\frac{\sigma^2 rn \log n}{m}}. \quad (69)$$

Combining the above two inequalities with (67) leads to

$$\eta^2 \geq \frac{1}{m}\|\mathcal{A}(M^\star - XX^T)\|^2 - C\|M^\star - XX^T\|_F \sqrt{\frac{\sigma^2 rn \log n}{m}} - \sqrt{\frac{\sigma^2 \log n}{m}} - |\sigma^2 - \hat{\sigma}^2|$$

$$\geq (1 - \delta)\|XX^T - M^\star\|_F^2 - C\|XX^T - M^\star\|_F \sqrt{\frac{\sigma^2 rn \log n}{m}} - \sqrt{\frac{\sigma^2 \log n}{m}} - |\sigma^2 - \hat{\sigma}^2|.$$

$$(70)$$

Now assuming that

$$\|XX^T - M^\star\|_F^2 \geq \max\left\{16C^2\frac{\sigma^2 rn\log n}{m}, 4\sqrt{\frac{\sigma^2\log n}{m}}, 4|\sigma^2 - \hat\sigma^2|\right\},$$

the inequality (70) can be further lower bounded as

$$\eta^2 \geq (1/4 - \delta)\|XX^T - M^\star\|_F^2 \geq \frac{1/4 - \delta}{1 + \delta}\frac{1}{m}\|\mathcal{A}(XX^T - M^\star)\|,$$

which completes the proof for the lower bound. The upper bound on $\eta^2$ can be established in a similar fashion. □

Now we are ready to prove Theorem 8.

*Proof.* We consider two cases. First, suppose that

$$\min_k \eta_k \lesssim \max\left\{\frac{\sigma^2 rn\log n}{m}, \sqrt{\frac{\sigma^2\log n}{m}}, |\sigma^2 - \hat\sigma^2|\right\}.$$

Combined with (70), this implies that

$$(1 - \delta)\|X_{k^*}X_{k^*}^T - M^\star\|_F^2 - C\|X_{k^*}X_{k^*}^T - M^\star\|_F\sqrt{\frac{\sigma^2 rn\log n}{m}}$$

$$\lesssim \max\left\{\frac{\sigma^2 rn\log n}{m}, \sqrt{\frac{\sigma^2\log n}{m}}, |\sigma^2 - \hat\sigma^2|\right\}. \tag{71}$$

Now, if $\|X_{k^*}X_{k^*}^T - M^\star\|_F \leq 2C\sqrt{\frac{\sigma^2 rn\log n}{m}}$, then the proof is complete. Therefore, suppose that $\|X_{k^*}X_{k^*}^T - M^\star\|_F > 2C\sqrt{\frac{\sigma^2 rn\log n}{m}}$. This together with (71) leads to

$$\|X_{k^*}X_{k^*}^T - M^\star\|_F^2 \lesssim \frac{1}{1/2 - \delta}\max\left\{\frac{\sigma^2 rn\log n}{m}, \sqrt{\frac{\sigma^2\log n}{m}}, |\sigma^2 - \hat\sigma^2|\right\},$$

which again completes the proof. Finally, suppose that

$$\min_k \eta_k \gtrsim \max\left\{\frac{\sigma^2 rn\log n}{m}, \sqrt{\frac{\sigma^2\log n}{m}}, |\sigma^2 - \hat\sigma^2|\right\}.$$

This combined with (67) implies that

$$(1 + \delta)\|X_{k^*}X_{k^*}^T - M^\star\|_F^2 + C\|X_{k^*}X_{k^*}^T - M^\star\|_F\sqrt{\frac{\sigma^2 rn\log n}{m}}$$

$$\gtrsim \max\left\{\frac{\sigma^2 rn\log n}{m}, \sqrt{\frac{\sigma^2\log n}{m}}, |\sigma^2 - \hat\sigma^2|\right\},$$

for every $k = 0, 1, \ldots, K$. If $\|X_{k^*}X_{k^*}^T - M^\star\|_F \leq 2C\sqrt{\frac{\sigma^2 rn\log n}{m}}$, then the proof is complete. Therefore, suppose that $\|X_{k^*}X_{k^*}^T - M^\star\|_F > 2C\sqrt{\frac{\sigma^2 rn\log n}{m}}$. This together with the above inequality results in

$$\|X_kX_k^T - M^\star\|_F^2 \gtrsim \frac{1}{3/2 + \delta}\max\left\{\frac{\sigma^2 rn\log n}{m}, \sqrt{\frac{\sigma^2\log n}{m}}, |\sigma^2 - \hat\sigma^2|\right\}$$

$$\gtrsim \max\left\{\frac{\sigma^2 rn\log n}{m}, \sqrt{\frac{\sigma^2\log n}{m}}, |\sigma^2 - \hat\sigma^2|\right\}$$

for every $k = 0, 1, \ldots, K$. Therefore, Lemma 22 can be invoked to show that

$$\eta_k \asymp \frac{1}{\sqrt{m}}\|\mathcal{A}(X_kX_k^T - M^\star)\|.$$

With this choice of $\eta_k$, the rest of the proof is identical to that of Theorem 7, and omitted for brevity. □

# H  Proof for Spectral Initialization (Proposition 6)

In this section we prove that spectral initialization is able to generate a sufficiently good initial point so that PrecGD achieves a linear convergence rate, even in the noisy case. For convenience we restate our result below.

**Proposition 23** (Spectral Initialization). *Suppose that $\delta \leq (8\kappa\sqrt{r^\star})^{-1}$ and $m \gtrsim \frac{1+\delta}{1-\delta}\frac{\sigma^2 rn \log n}{\rho^2 \lambda_{r^\star}^2(M^\star)}$ where $\kappa = \lambda_1(M^\star)/\lambda_{r^\star}(M^\star)$. Then, with high probability, the initial point $X_0$ produced by (18) satisfies the radius condition (17).*

*Proof.* Let $\mathcal{A}^* : \mathbb{R}^m \rightarrow \mathbb{R}^{n \times n}$ be the dual of the linear operator $\mathcal{A}(\cdot)$, defined as $\mathcal{A}^*(y) = \sum_{i=1}^{m} y_i A_i$. Based on this definition, the initial point $X_0 \in \mathbb{R}^{n \times r}$ satisfies $X_0 = \mathcal{P}_r\left(\frac{1}{m}\mathcal{A}^*(y)\right)$, where we recall that

$$\mathcal{P}_r(M) = \arg\min_{X \in \mathbb{R}^{n \times r}} \|XX^T - M\|_F.$$

Define $E = X_0 X_0^T - M^\star$, and note that $\text{rank}(E) \leq 2r$. It follows that

$$
\begin{aligned}
\|E\|_F &= \sqrt{\sum_{i=1}^{r} \sigma_i(E)^2 + \sum_{i=r+1}^{2r} \sigma_i(E)^2} \leq \sqrt{2}\|E\|_{F,2r} \\
&\leq \sqrt{2}\left\|X_0 X_0^T - \frac{1}{m}\mathcal{A}^*(y)\right\|_{F,2r} + \sqrt{2}\left\|\frac{1}{m}\mathcal{A}^*(y) - M^\star\right\|_{F,2r} \\
&\leq 2\sqrt{2}\left\|\frac{1}{m}\mathcal{A}^*(y) - M^\star\right\|_{F,2r} \\
&\leq 2\sqrt{2}\left\|\frac{1}{m}\mathcal{A}^*(\mathcal{A}(M^\star)) - M^\star\right\|_{F,2r} + 2\sqrt{2}\left\|\frac{1}{m}A_i \epsilon_i\right\|_{F,2r} \\
&\leq 2\sqrt{2}\delta\|M^\star\|_F + 2\sqrt{2}\left\|\frac{1}{m}A_i \epsilon_i\right\|_{F,2r}.
\end{aligned}
$$

Now, note that $\|M^\star\|_F \leq \sqrt{r^*}\kappa\lambda_{r^*}(M^\star)$. Moreover, due to Lemma 16, we have

$$2\sqrt{2}\left\|\frac{1}{m}A_i \epsilon_i\right\|_{F,2r} \leq 2\sqrt{2}\sqrt{2r}\left\|\frac{1}{m}A_i \epsilon_i\right\|_2 \lesssim \sqrt{\frac{\sigma^2 rn \log n}{m}}. \tag{72}$$

This implies that

$$\frac{1}{m}\|\mathcal{A}(X_0 X_0^T - M^\star)\|^2 \leq 16(1+\delta)r^*\kappa^2\lambda_{r^*}(M^\star)^2\delta^2 + C\frac{\sigma^2 rn \log n}{m}$$

Therefore, upon choosing $\delta \leq \frac{\rho}{8\sqrt{r^*}\kappa}$ and $m \gtrsim \frac{1+\delta}{1-\delta}\frac{\sigma^2 rn \log n}{\rho^2 \lambda_{r^*}^2(M^\star)}$, we have

$$\frac{1}{m}\|\mathcal{A}(XX^T - M^*)\|^2 \leq \rho^2(1-\delta)\lambda_{r^*}(M^\star)^2 \tag{73}$$

This completes the proof. □

# I  Proof of Lemma 16

First we state a standard concentration inequality. A proof of this result can be found in Tropp [56].

**Lemma 24** (Matrix Bernstein's inequality). *Suppose that $\{W_i\}_{i=1}^{m}$ are matrix-valued random variables such that $\mathbb{E}[W_i] = 0$ and $\|W_i\|_2 \leq R^2$ for all $i = 1, \ldots, m$. Then*

$$\mathbb{P}\left(\left\|\sum_{i=1}^{m} W_i\right\| \geq t\right) \leq n\exp\left(\frac{-t^2}{2\left\|\sum_{i=1}^{m}\mathbb{E}[W_i^2]\right\|_2 + \frac{2R^2}{3}t}\right).$$

We also state a standard concentration bound for the operator norm of Gaussian ensembles. A simple proof can be found in Wainwright [57].

**Lemma 25.** *Let $A \in \mathbb{R}^{n \times n}$ be a standard Gaussian ensemble with i.i.d. entries. Then the largest singular value of $A$ (or equivalently, the operator norm) satisfies*

$$\sigma_{\max}(A) \leq (2 + c)\sqrt{n}$$

*with probability at least $1 - 2\exp(-nc^2/2)$.*

For simplicity, we assume that the measurement matrices $A_i, i = 1, \ldots m$ are fixed and all satisfy $\|A_i\| \leq C\sqrt{n}$. Due to Lemma 25, this assumption holds with high probability for Gaussian measurement ensembles. Next, we provide the proof of Lemma 16.

Proof of Lemma 16. First, note that $\|A_i \varepsilon_i\|_2 \leq \|A_i\| \cdot |\varepsilon_i|$. The assumption $\|A_i\| \lesssim \sqrt{n}$ implies that $\|A_i \varepsilon_i\|$ is sub-Gaussian with parameter $C\sqrt{n}\sigma$. Therefore, we have $\mathbb{P}(\|A_i \varepsilon\| \gtrsim \sqrt{n}t) \geq 1 - 2\exp\left(-\frac{t^2}{2\sigma^2}\right)$. Applying the union bound yields

$$\mathbb{P}(\max_{i=1,\ldots,m} \|A_i \varepsilon\| \geq \sqrt{n}t) \geq 1 - 2m\exp\left(-\frac{t^2}{2\sigma^2}\right).$$

Moreover, one can write

$$\left\| \sum_{i=1}^{m} \mathbb{E}[(A_i \varepsilon_i)^2] \right\| \leq \sum_{i=1}^{m} \|A_i\|^2 \mathbb{E}[\varepsilon_i^2] \lesssim \sigma^2 mn \tag{74}$$

Using Matrix Bernstein's inequality, we get

$$\mathbb{P}\left(\frac{1}{m} \left\| \sum_{i=1}^{m} A_i \varepsilon \right\| \leq t\right) \geq 1 - n\exp\left(-\frac{t^2 m^2}{2C\sigma^2 mn + \frac{2}{3}C'\sqrt{n}mt}\right) - 2m\exp\left(-\frac{t^2}{2}\right).$$

Using $t \asymp \sqrt{\frac{\sigma^2 n \log n}{m}}$ in the above inequality leads to

$$\mathbb{P}\left(\frac{1}{m} \left\| \sum_{i=1}^{m} A_i \varepsilon \right\| \lesssim \sqrt{\frac{\sigma^2 n \log n}{m}}\right) \geq 1 - n^{-C} - 2m\exp\left(-\frac{t^2}{2}\right)$$

$$\gtrsim 1 - 3n^{-C},$$

where the last inequality follows from the assumption $m \gtrsim \sigma n \log n$. This completes the proof. $\square$