# OpenReview forum: "Preconditioned Gradient Descent for Over-Parameterized Nonconvex Matrix Factorization"
_NeurIPS.cc/2021/Conference — NeurIPS 2021 Poster_

### Official Review · Reviewer_Yim3 · 2021-07-14

**Rating:** 5
**Confidence:** 3

**Summary:**

This paper considers the problem of matrix sensing in the over-parameterized setting where the rank of the estimator is larger than that of the ground truth. A new algorithm called preconditioned gradient descent is proposed, and its performance is discussed both theoretically and numerically.

**Limitations And Societal Impact:**

Yes

**Main Review:**

Overall, this paper is well-written and easy to read. The literature review is comprehensive, and the discussion about the limitation of prior art as well as the intuition of the proposed algorithm is very helpful.

My main concern is about the theoretical guarantee in the noisy setting (Theorem 8). The current error bound is not minimax optimal. It seems quite crude because authors simply adds up the three error terms that appear in the proof, and the sub-optimality might be attributed to the theoretical analysis.  It would improve the quality if authors can improve the error bound.

Typo:
Theorem 5 in Line 237 should be Corollary 5.



**Time Spent Reviewing:**

4

---

> ### Author Response · Authors · 2021-08-10
> **Response to reviewer Yim3**
>
> We thank the reviewer for the valuable feedback. Please see the following response to your comments.
> > My main concern is about the theoretical guarantee in the noisy setting (Theorem 8). The current error bound is not minimax optimal. It seems quite crude because authors simply adds up the three error terms that appear in the proof, and the sub-optimality might be attributed to the theoretical analysis. It would improve the quality if authors can improve the error bound.
>
> * We wish to point out that it is Theorem 7 that states the minimax optimality of our algorithm, under the suitable choice of the regularization parameter $\eta\_k\approx\|X\_k X\_k^T - M^\star\|\_F$ stated throughout the paper.
>
> * We do not claim that Theorem 8 is minimax optimal; in fact, it must be suboptimal in order for it to be correct. In the noisy case, the RIP-based estimate of $\eta\_k\approx\|X\_k X\_k^T - M^\star\|\_F$ is biased, so we propose correcting this bias using an estimate of the noise variance. This works well in practice, but the error must necessarily be suboptimal. First, the empirical variance $\frac 1 m \sum\_{i=1}^m \varepsilon\_i$ must have a deviation from its expectation $\sigma$ (captured by the term $\mathcal{E}\_{dev}$). Second, if the variance estimate is inaccurate, then this contributes an additional source of suboptimality (captured by the term $\mathcal{E}\_{var}$). Both sources of suboptimality are fundamental to this bias-compensation scheme, and cannot be eliminated by a more careful analysis.

---

### Official Review · Reviewer_KAr6 · 2021-07-16

**Rating:** 8
**Confidence:** 4

**Summary:**

The authors consider low-rank matrix recovery problems, where the $n\times n$ matrix variable $M$ is factorized as $M=XX^T$, for some $X$ of size $n\times r$, with $r\ll n$. It is known from previous work that, although they are typically non-convex, these problems can oftentimes be solved through simple gradient descent over $X$, applied to a suitable loss function.

However, gradient descent tends to converge very slowly when either the ground true matrix is ill-conditioned, or the problem is overparameterized (that is, $r$ is strictly larger than the rank of the ground true matrix). In Reference [11], Tong, Ma and Chi have proposed a gradient preconditioner which solves the slow convergence issue when it is due to ill-conditioning. However, the method does not work when the problem is overparameterized.

In this article, the authors propose a refined preconditioner, which accommodates both ill-conditioning and overparameterization. In the particular setting of RIP matrix sensing with least-squares cost, they show that, with this preconditioner, gradient descent converges linearly in a neighborhood of the solution, up to the unavoidable statistical error.

**Limitations And Societal Impact:**

Not applicable to this article.

**Main Review:**

I liked this article.
* It is well-written and pleasant to read.
* I did unfortunately not find the time to read all proofs, but I read everything related to the noiseless case (until page 22 of the supplementary material), and found no issue.
* The subject of the article is important: slow convergence of gradient descent in overparameterized settings is a significant obstacle for successfully using Burer-Monteiro methods in practice. Proposing an algorithm which is capable of overcoming this issue is therefore, in my opinion, a very valuable contribution, although its significance is limited by the fact that it is unclear to me whether the algorithm applies to other problems than RIP matrix sensing (see Remark 1 below).
* The principle of the algorithm looks quite simple (the main idea is to add a very classical regularizer to the ScaledGD preconditioner), but proving that it works is rather involved.

I have the following points of criticism.
1. The fact that the damping parameter must satisfy Equation (1) looks like a rather strong constraint. The article explains how to find $\eta_k$ satisfying this equation in the case of RIP matrix sensing, but RIP matrix sensing is a rather particular problem. I think it would be important to discuss whether this constraint prevents the algorithm from being applicable to other problems (which would significantly reduce its interest), or whether (and how) it can be overcome.
2. Some clarifications in the setting would be welcome. In particular, the assumptions on the noise $\epsilon$ should be explicitely stated at the beginning (otherwise, it is difficult to understand Proposition 6, for instance). I also did not find in the article the assumption $||A_i|| \lesssim \sqrt{n}$ used in the proof of Lemma 17.
3. It seems to me that the proof of Lemma 14 can be significantly simplified (see the end of the review for a proposition of a simpler proof).
4. I find Section 7 (Numerical experiments) a bit too short and confusing.
   * Why did the authors choose the setting of their experiment as in Example 12 of reference [13]? The corresponding data matrices are very peculiar.
   * A single example (although three different cost functions are considered, the underlying sensing operator is the same in each case), moreover in very small dimension, is not enough to establish that PrecGD works better than GD and ScaledGD.
   * It seems that PrecGD stagnates: the error does not go to zero. This is especially visible on the figures of the first column. What happens?
   * The experiment entangles three sources of difficulty: the ill-conditioning of the ground truth, the over-parameterization and the non-smoothness of the objective function. As a consequence, it is difficult to tell what exactly makes GD and ScaledGD fail or converge slowly. The fact that the figures in the first and second row of Figure 2 are quite similar gives the impression that the issues encountered by ScaledGD are not caused by overparameterization. Therefore, it does not illustrate the main message of the article (which is that PrecGD works better than ScaledGD when the problem is overparameterized).
   * If PrecGD is able to successfully handle non-smooth objectives whereas GD and ScaledGD fail, this is very interesting, but also a bit mysterious. Would it be possible to provide at least an intuitive explanation?
5. There are many typos in Appendix D.1. I recommend proofreading this subsection.

In addition, I have the following minor remarks.
* Line 27: "exists" $\to$ "exist"?
* Line 59: "suggests" $\to$ "suggest"
* Line 89: "logarithimic" $\to$ "logarithmic"
* Line 90: space missing before "$\lambda_1$"
* Line 110: $\sigma$ has not been previously defined.
* The assertion at lines 118-119 is not correct. From my understanding of [31], the strict saddle property also holds in the over-parameterized regime. Indeed, Theorem 1.2 of [31] shows that, even in the over-parameterized regime, if the RIP constant is smaller than some threshold, then for any non globally optimal $X$, either $\nabla f(X)\ne 0$ or $\nabla^2 f(X)\not\succeq 0$, which is to say that all non-optimal points have a descent direction.
* Equation (4): "$f^*=\min f(X)$" $\to$ "$f^*=\min f$" or "$f^*=\min_X f(X)$".
* Equation after line 164: I think there should be a factor "$2$" before "$\alpha$".
* Line 165: the value of $f(X_{k+1})$ is not correct.
* Line 184: "forces" $\to$ "force".
* Line 188: "$\nabla f(X)(X^TX+\eta I)$" $\to$ "$\nabla f(X)(X^TX+\eta I)^{-1}$"
* Lines 196 and 531, and Equation (32): "$\eta$" $\to$ "$\eta I$".
* Equation (25): there should be no point at the end.
* Line 208: "the the" $\to$ "the".
* Line 265: "Candes" $\to$ "Candès"; "any other estimator" $\to$ "any estimator"?
* Line 289: "$f(X)$" $\to$ "$f(X_k)$".
* Equation (35): I think there should be no sum over $i$.
* Between Lemmas 12 and 13: in my opinion, it should be explicitely said at the beginning that this is a high-level description of the proof, not a rigorous proof. In addition, this high-level description presents several issues because of which I could not understand it:
  * Equation after line 583: I think the right-hand side is not correct (a scalar product is missing).
  * Equation (37): the inequality is not correct. First, there is no reason why $\sum_{i=k+1}^r \lambda_i^2(X^TX)$ should be smaller than $\lambda_{k+1}^2(X^TX)$. Second, Lemma 13 does not prove that $||Z^T(I-U_kU_k^T)Z||_F \lesssim ||E||_F^2$: it proves that $||Z^T(I-U_kU_k^T)Z||_F \lesssim ||E||_F$ (without the square).
  * If $||E||\_F^2 \geq 1 $, then Equation (37) does not imply that $\lambda_{k+1}(X^TX)$ is large whenever $\sin(\theta_k)$ is large.
  * Line 593: what is $T_1$?
* Equations (44) and (45): "$\lambda_{k+1}^2(X^TX)$" should be "$\lambda_{k+1}(X^TX)$".
* I found it a bit difficult to understand the induction in the proof of Theorem 4; I recommend stating the inductive assumption explicitely.
* Line 625: "where we prove" $\to$ "that is, we prove"?
* Second line of Equation (46): there is a factor $2$ missing.
* Lines 641-642: what does "the rank can be verified" mean?
* Line 645: the definition of $d$ is not correct (there is a mismatch between the dimensions of $\Sigma_k^{-1}$ and the size of $e$); it should be $d=V_k\Sigma_k^{-1}U_k^T$.
* Problem after line 673, first line: there should be parentheses around $(1+\delta)\mathop{Tr}(V)-(1-\delta)\mathop{Tr}(U)$.
* Line 674: I think that there should be a division by $||w||$ in the equality which defines $M(w)$.
* Problem after line 674, first line: there should be no "$(1+\delta)$".
* Line 679: there are traces missing.
* Line 691: if we assume that $k=r^*$, then $X_1,Z_1$ belong to $\mathbb{R}^{r\_* \times r\_* }$.
* Equation after line 724: in the second line, I think the "$||Z||\_F$" should be replaced with "$||Z_2|\|_F$" (without this replacement the second inequality has no chance to hold when $Z_2=0$).

Simpler proof of Lemma 14: Let $Y$ maximize the right-hand side of Equation (47). Let $W$ be the matrix corresponding to the orthogonal projection onto $\mathrm{Range}(X)+\mathrm{Range}(Z)$. If we set $\tilde Y=W Y$, it holds that
$$\langle X\tilde Y^T + \tilde Y X^T, E\rangle
 = \langle X Y^T, E W\rangle + \langle Y X^T, W E\rangle
 = \langle X Y^T + Y X^T, E\rangle$$.
On the other hand,
$$ ||X\tilde Y^T+\tilde Y X^T||_F = ||W (XY^T+YX^T) W||_F \leq ||XY^T+YX^T||_F$$
and $||\tilde Y||_P = ||WYP||_F \leq ||YP||_F = ||Y||_P\leq 1$.
This allows to show that $\tilde Y$ also maximizes the right-hand side of Equation (47).

Since $\mathrm{Range}(\tilde Y)\subset \mathrm{Range}(X)+\mathrm{Range}(Z)$, the $(2r,\delta)$-RIP implies
$$
|\langle \mathcal{A}(X\tilde Y^T+\tilde Y X^T),\mathcal{A}(E)\rangle -
\langle X\tilde Y^T+\tilde Y X^T,E\rangle| \leq
\delta ||X \tilde Y^T + \tilde Y X^T||\\,||E||.
$$
As a consequence,
$$ \begin{align*}
||\nabla f(X)||_{P^*}
& = \underset{||Y||\_P \leq 1}{\max} \langle \mathcal{A}(XY^T+YX^T),\mathcal{A}(E)\rangle \\\\
& \geq \langle \mathcal{A}(X\tilde Y^T+\tilde Y X^T),\mathcal{A}(E)\rangle \\\\
& \geq \langle X\tilde Y^T+\tilde Y X^T,E\rangle - \delta ||X \tilde Y^T + \tilde Y X^T||\\,||E|| \\\\
& = \underset{||Y||_P \leq 1}{\max} \langle X Y^T+ Y X^T,E\rangle - \delta ||X Y^T + Y X^T||\\,||E||.
\end{align*}$$

**Added after author feedback**

I thank the authors for adequately addressing my concerns, as well as for the discussion we had during the discussion period. I have increased my grade and hope the article will be accepted.

**Time Spent Reviewing:**

8

---

> ### Author Response · Authors · 2021-08-10
> **Response to reviewer KAr6**
>
> First we would like to thank the reviewer for such a careful read of our paper and also for the extremely insightful feedback. We will fix all the typos pointed out by the reviewer. Please see our response below.
>
> > The fact that the damping parameter must satisfy Equation (1) looks like a rather strong constraint. The article explains how to find $\eta_k$ satisfying this equation in the case of RIP matrix sensing, but RIP matrix sensing is a rather particular problem. I think it would be important to discuss whether this constraint prevents the algorithm from being applicable to other problems (which would significantly reduce its interest), or whether (and how) it can be overcome.
>
> * While the theoretical analysis in our paper uses some properties specific to RIP matrix sensing, we believe our results can be extended to similar problems such as matrix competition and robust PCA. In the original ScaledGD paper of Tong et al. [43], the authors use matrix sensing as a starting point, but later extend their results to matrix completion and robust PCA. The reason they are able to do this is because the incoherence assumption behind these problems is very similar in principle to RIP. Likewise, we believe that incoherence can be used to select $\eta_k$ with the desired properties, so that PrecGD converges linearly for these problems as well. Indeed, in our own numerical experiments, we observe that choosing $\eta_k$ by replacing RIP with incoherence works similarly well. Incoherence is not the focus of this particular work, though it is an important follow-up direction.
>
> > Some clarifications in the setting would be welcome. In particular, the assumptions on the noise  should be explicitly stated at the beginning (otherwise, it is difficult to understand Proposition 6, for instance). I also did not find in the article the assumption $\|A_i\|\lesssim \sqrt{n}$.
>  used in the proof of Lemma 17.
>
> *  Thanks again for carefully reading our paper and catching these small errors. We will fix them in the updated version.
>
> > It seems to me that the proof of Lemma 14 can be significantly simplified (see the end of the review for a proposition of a simpler proof).
>
> * The reviewer is completely right. The provided proof uses the same idea, namely performing an orthogonal projection onto the range of $X$ and $Z$, but its argument is simpler and much easier to understand. We will use this version of the proof in our updated version. We express our gratitude to the reviewer for this insightful remark and will certainly acknowledge the reviewer in the new version.
>
> > Why did the authors choose the setting of their experiment as in Example 12 of reference [13]? The corresponding data matrices are very peculiar.
> * We have also experimented with Gaussian measurements and the results are almost identical. However, within the context of RIP matrix ensembles, Gaussian matrices are well-known to be "easier" choices. We chose the specific measurement operator in Zhang et al. [13] as it is the "hardest" choice, in the sense that it has the smallest RIP constant among all measurement operators with spurious local minima. By contrast, a Gaussian operator with the same RIP constant would likely not have spurious local minima, and is in this sense "easier".
>
> > It seems that PrecGD stagnates: the error does not go to zero. This is especially visible on the figures of the first column. What happens?
> * See also our response to reviewer hWq7. In our numerical experiments for non-smooth functions, we used a constant step-size in order to compare the performance of GD and PrecGD. In general, for non-smooth functions (that do not satisfy gradient Lipschitzness by construction), a diminishing step-size is necessary for avoiding stagnation. This has been demonstrated rigorously for ScaledGD (see [A1]). We believe that similar conditions hold for PrecGD. However, understanding the performance of PrecGD in a non-smooth setting is not the main focus of this paper; it is left as future work.
>
> > The numerical experiments entangles three sources of difficulty and it is not clear PrecGD works better than ScaledGD when the problem is over-parameterized.
>
> * We agree with the reviewer's comments here. From figure 2, it is not easy to see our main message, i.e., PrecGD works better than ScaledGD when the problem is over-parameterized.  Our goal in this section was to show that PrecGD works surprisingly well even in the non-smooth setting, a phenomenon that we do not yet fully understand. We agree that the numerical results section should put more emphasis on our main message. We will make changes to the numerical results section to reflect these problems.
>
> * Moreover, we would like to mention that when the problem has exact parameterization, ScaledGD has also been shown to work well in the non-smooth setting with a very specific step-size choice (see [A1]). Therefore, we do believe that in figure 2, the failure of ScaledGD is caused by over-parameterization instead of non-smoothness, while PrecGD successfully overcomes the difficulties caused by over-parameterization.
>
> > If PrecGD is able to successfully handle non-smooth objectives whereas GD and ScaledGD fail, this is very interesting, but also a bit mysterious. Would it be possible to provide at least an intuitive explanation?
>
> * In the non-smooth setting, the failure of GD is easy to explain. If $p<2$ in the $\ell_p$ objective function, the problem becomes locally weakly convex, so GD is just unable to converge at a linear rate, regardless of whether the problem is over-parameterized.
>
> * In the case with exact parameterization $r=r^*$, ScaledGD has actually been shown to converge linearly in the non-smooth setting, although a very specific choice of the step-size is required. (See [A1].) However, this guarantee does fail in the over-parameterized case $r>r^*$, for the same reasons as the smooth case.
>
> * In figure 2, we see that PrecGD is able to converge in the non-smooth, over-parameterized setting, with just a constant step-size. In general, $\ell_2$ regularization is known to have a smoothing effect on the objective. We have already shown that adding an $\ell_2$ regularizer makes the problem Lipschitz smooth (under the $P$-norm), and as a result, PrecGD overcomes the sporadic behavior of ScaledGD in the over-parameterized regime. We suspect that $\ell_2$ regularization also provides a smoothing effect for non-smooth objectives (that do not satisfy gradient Lipschitzness by construction) under the $P$-norm. We think this is the main reason the PrecGD works well in the  non-smooth cases, although a rigorous analysis is left for future work.
>
> > The assertion at lines $118-119$ is not correct. From my understanding of [31], the strict saddle property also holds in the over-parameterized regime. Indeed, Theorem $1.2$ of [31] shows that, even in the over-parameterized regime, if the RIP constant is smaller than some threshold, then for any non globally optimal $X$, either $\nabla f(X) \neq 0$ or $\nabla^{2} f(X) \nsucceq 0$, which is to say that all non-optimal points have a descent direction.
>
> * We truly appreciate the reviewer's insightful comment. Note that, in order to satisfy strict saddle property for an objective function $f(x)$, one of the following conditions must be satisfied for any feasible solution $x$:
>  1.  the norm of the gradient must be sufficiently large, i.e., $\|\nabla f(x)\|\geq \epsilon_g$, for some $\epsilon_g>0$;
>  2.  the Hessian must have a sufficiently negative curvature, i.e., $\lambda_{\min}(\nabla^2 f(x))\leq -\epsilon_H$, for some $\epsilon_H>0$;
>  3. the point must lie close to a globally optimal solution.
>
> * A typical proof uses the violation of Conditions 1 and 2 to imply Condition 3, which justifies near-global optimality.
> As the reviewer correctly pointed out, in the context of over-parameterized matrix factorization, any non-global solution must either satisfy $\nabla f(x)\not=0$ or $\lambda\_{\min}(\nabla^2 f(x))<0$. But in the over-parameterized regime, a violation of Conditions 1 and 2 does not necessarily imply Condition 3. Intuitively, the landscape about the ground truth is "flat" in the "redundant" directions introduced by over-parameterization. Therefore, over-parameterized matrix factorization may not satisfy the strict saddle property.
>
> [A1] Tong, Tian, Cong Ma, and Yuejie Chi. "Low-rank matrix recovery with scaled subgradient methods: Fast and robust convergence without the condition number." IEEE Transactions on Signal Processing 69 (2021): 2396-2409.
>
> [13] Zhang, Richard Y., Somayeh Sojoudi, and Javad Lavaei. "Sharp Restricted Isometry Bounds for the Inexistence of Spurious Local Minima in Nonconvex Matrix Recovery." J. Mach. Learn. Res. 20.114 (2019): 1-34.
>
> [43] Tong Tian, Cong Ma, and Yuejie Chi. "Accelerating ill-conditioned low-rank matrix estimation via scaled gradient descent." arXiv preprint arXiv:2005.08898 (2020).

---

> ### Author Response · Authors · 2021-08-10
> **Response to reviewer KAr6 (continued)**
>
> > Between Lemmas 12 and 13: in my opinion, it should be explicitely said at the beginning that this is a high-level description of the proof, not a rigorous proof. In addition, this high-level description presents several issues because of which I could not understand it:
> >* Equation after line 583: I think the right-hand side is not correct (a scalar product is missing).
> >*  Equation (37): the inequality is not correct. First, there is no reason why $\sum\_{i=k+1}^{r} \lambda\_{i}^{2}\left(X^{T} X\right)$ should be smaller than $\lambda_{k+1}^{2}\left(X^{T} X\right)$.
> Second, Lemma 13 does not prove that $\left\|Z^{T}\left(I-U\_{k} U\_{k}^{T}\right) Z\right\|\_{F} \lesssim\|E\|\_{F}^{2}:$ it proves that $\left\|Z^{T}\left(I-U\_{k} U\_{k}^{T}\right) Z\right\|_{F} \lesssim\|E\|\_{F}$ (without the square).
> >* If $\|E\|\_{F}^{2} \geq 1$, then Equation (37) does not imply that $\lambda\_{k+1}\left(X^{T} X\right)$ is large whenever $\sin \left(\theta_{k}\right)$ is large.
>
> We agree with all the reviewer's comments, which have pointed a bug
> in our original proof (the missing scalar product). Here we provide
> a full rewrite of these arguments, while fixing the bugs.
>
> Recall that
> we have $XX^{T}=U\_{k}\Lambda_{k}U\_{k}^{T}+R$ where $U\_{k}^{T}R=0$.
> We put the top $k$ eigenvectors of $XX^{T}$ in $U\_{k}\Lambda\_{k}U\_{k}^{T}$
> and the remaining eigenvectors in $R$.
>
> Write $\Pi\_{k}=I-U\_{k}U\_{k}^{T}$
> and note that
> $$
> \Pi_{k}(XX^{T}-M^{\star})\Pi\_{k}=\Pi\_{k}XX^{T}\Pi\_{k}-\Pi\_{k}M^{\star}\Pi\_{k}=R-\Pi\_{k}M^{\star}\Pi\_{k}.
> $$
> This yields by subaddivity of the norm
> $$
> \|R-\Pi\_{k}M^{\star}\Pi\_{k}\|\_{F}\le\|R\|\_{F}+\|\Pi\_{k}M^{\star}\Pi\_{k}\|\_{F}.
> $$
> Dividing by $\|E\|\_{F}$ recovers the definition of $\sin\theta\_{k}$
> and an upper-bound
> $$
> \frac{\|R-\Pi\_{k}M^{\star}\Pi\_{k}\|\_{F}}{\|E\|\_{F}}=\sin\theta\_{k}\le\frac{\|\Pi\_{k}M^{\star}\Pi\_{k}\|\_{F}}{\|E\|\_{F}}+\frac{\|R\|\_{F}}{\|E\|\_{F}}.
> $$
> In Lemma 13, we prove for $\rho$ satisfying $\|E\|\_{F}/\lambda\_{\min}(Z^{T}Z)\le\rho\le1/\sqrt{2}$
> that
> $$
> \frac{\|\Pi\_{k}M^{\star}\Pi\_{k}\|\_{F}}{\|E\|\_{F}}\le\frac{1}{\sqrt{2}}\frac{\rho}{\sqrt{1-\rho^{2}}}\le\rho.
> $$
> Also, we trivially have
> $$
> \|R\|\_{F}\le\|R\|\cdot\sqrt{\mathrm{rank}(R)}=\lambda\_{k+1}(XX^{T})\cdot\sqrt{r-k}.
> $$
> Substituting both results and $\lambda\_{k+1}(XX^{T})=\lambda\_{k+1}(X^{T}X)$
> yields
> $$
> \sin\theta\_{k}\le\rho+\sqrt{r-k}\cdot\frac{\lambda\_{k+1}(X^{T}X)}{\|E\|\_{F}}.
> $$
> This is the key bound for our inductive argument. Recall that by Lemma
> 12, we need to demonstrate that $\lambda\_{k}(X^{T}X)$ is large and
> $\sin\theta\_{k}$ is small for the same value of $k$, in order to
> establish a lower-bound on $\mu\_{P}$. We begin with the base case.
> If $k=r^{*}$, then by Weyl's inequality on $\|E\|\_{F}/\lambda\_{\min}(Z^{T}Z)\le\rho$,
> we have $\lambda\_{k}(X^{T}X)$ large. If $\sin\theta_{k}$ is also
> small, then we are done. But if $\sin\theta\_{k}$ is large, then the
> largeness of $\sin\theta\_{k}$ gives a lower-bound on $\lambda\_{k+1}(X^{T}X)$
> via the inequality above. Then we repeat the above by induction, arguing
> that if $\sin\theta\_{k+1}$ is large, then it implies a lower-bound
> on $\lambda\_{k+2}(X^{T}X)$, and so on. Finally, we arrive at $k=r$. At this point, $\sin\theta\_{k}$ is guaranteed to be small via Lemma
> 13, and we are done.
>
> The proof of Theorem 4 fills in the details on
> this argument, using $\mu$ as a threshold between a "large" and
> "small" value of $\sin\theta\_{k}$. We promise to revise this
> section carefully in the paper, and acknowledge the reviewer accordingly.

---

> > ### Comment · Reviewer_KAr6 · 2021-08-13
> > **Two questions on author feedback**
> >
> > Dear authors,
> >
> > Thank you very much for your feedback. I have read it carefully, and have two questions / comments about it.
> >
> > 1) You say
> > > Moreover, we would like to mention that when the problem has exact parameterization, ScaledGD has also been shown to work well in the non-smooth setting with a very specific step-size choice (see [A1]). Therefore, we do believe that in figure 2, the failure of ScaledGD is caused by over-parameterization instead of non-smoothness, while PrecGD successfully overcomes the difficulties caused by over-parameterization.
> >
> >     But the first line of Figure 2 presents experiments done in the exactly parameterized setting. And except when $p=1.7$, ScaledGD seems to fail, which, from my understanding, is in contradiction with your reply. Where does this contradiction come from? Is it because you use a bad stepsize? If yes, I think the experiments should be redone with a better stepsize.
> >
> > 2) I disagree with the following.
> > > > The assertion at lines $118 - 119$ is not correct. From my understanding of [31], the strict saddle property also holds in the over-parameterized regime. Indeed, Theorem $1.2$ of [31] shows that, even in the over-parameterized regime, if the RIP constant is smaller than some threshold, then for any non globally optimal $X$, either $\nabla f(X)\ne 0$ or $\nabla^2 f(X) \not \succeq 0$ , which is to say that all non-optimal points have a descent direction.
> >
> >     > * We truly appreciate the reviewer's insightful comment. Note that, in order to satisfy strict saddle property for an objective function $f(x)$, one of the following conditions must be satisfied for any feasible solution $x$:
> >     >   1) the norm of the gradient must be sufficiently large, i.e., $|\nabla f(x)| \geq \epsilon_g$, for some $\epsilon_g>0$;
> >     >   2) the Hessian must have a sufficiently negative curvature, i.e., $\lambda_{\min}(\nabla^2 f(x)) \leq -\epsilon_H$, for some $\epsilon_H>0$;
> >     >   3) the point must lie close to a globally optimal solution.
> >
> >     > * A typical proof uses the violation of Conditions 1 and 2 to imply Condition 3, which justifies near-global optimality. As the reviewer correctly pointed out, in the context of over-parameterized matrix factorization, any non-global solution must either satisfy $\nabla f(x)$ or $\lambda_{\min}(\nabla^2 f(x))<0$. But in the over-parameterized regime, a violation of Conditions 1 and 2 does not necessarily imply Condition 3. Intuitively, the landscape about the ground truth is "flat" in the "redundant" directions introduced by over-parameterization. Therefore, over-parameterized matrix factorization may not satisfy the strict saddle property.
> >
> >     Indeed, let $\eta>0$ be arbitrary. Let us define $E_{\eta}$ the $\eta$-neighborhood of the set of global minimizers of $f$, that is
> >     $$E_{\eta} = \\{ X\in\mathbb{R}^{n\times r}, d(X,\mathrm{argmin} f) < \eta \\}.$$
> >     For any $n>0$, we set
> >     $$D_n = \\{ X \in \mathbb{R}^{n\times r} - E_{\eta} \mbox{ such that } ||\nabla f(X)|| \leq \frac{1}{n} \mbox{ and } \lambda_{\min}(\nabla^2 f(X)) \geq - \frac{1}{n} \\}. $$
> >     One can check that, under the RIP assumption, $||\nabla f(X)|| \to \infty$ when $||X|| \to \infty$. Therefore, there exists $R>0$ such that, for all $n$, $D_n \subset \overline{B(0,R)}$. (Here, $\overline{B(0,R)}$ is the closed Euclidean ball of radius $R$ in $\mathbb{R}^{n\times r}$.)
> >
> >     Therefore, $(D_n)\_{ n \in \mathbb{Z}\_{>0}}$ is a sequence of closed subsets of the compact set $\overline{B(0,R)}$. For any $n$, we have $D_{n+1} \subset D_n$. From Theorem $2.1$ of [31], the intersection of all $D_n$ is empty. From the definition of compactness, there must therefore exist some $N$ such that $D_N=\emptyset$.
> >
> >     If we set $\epsilon_g = \epsilon_H = \frac{1}{N}$, then any $X\in\mathbb{R}^{n\times r}$ must satisfy one of the following three conditions:
> >       1) the norm of the gradient is sufficiently large, i.e., $||\nabla f(X)|| \geq \epsilon_g$;
> >       2) the Hessian has a sufficiently negative curvature, i.e., $\lambda_{\min}(\nabla^2 f(X)) \leq -\epsilon_H$;
> >       3) the point is $\eta$-close to a globally optimal solution. (Recall that $\eta$ can be as small as desired.)
> >
> >     Therefore, $f$ satisfies the strict saddle property.

---

> > > ### Author Response · Authors · 2021-08-17
> > > **Response to additional questions**
> > >
> > > We thank the reviewer again for the very insightful comments. Please see our response below.
> > >
> > > > But the first line of Figure 2 presents experiments done in the exactly parameterized setting. And except when $p=1.7$, ScaledGD seems to fail, which, from my understanding, is in contradiction with your reply. Where does this contradiction come from? Is it because you use a bad stepsize? If yes, I think the experiments should be redone with a better stepsize.
> > >
> > > * In Figure 2, we used a constant step-size. We agree with the reviewer that this makes it unclear whether the failure of ScaledGD is due to over-parameterization or non-smoothness. In our own experiments, ScaledGD continues to fail even with the specific step-sizes given in Tong et al. [R1] for the non-smooth, exactly parameterized case. We will update Figure 2 to show that the failure of ScaledGD is indeed due to over-parameterization.
> > >
> > > > I disagree with the following...Therefore, $f$ satisfies the strict saddle property.
> > >
> > > * We thank the reviewer for this insightful comment. We would like to make some clarifications regarding our previous response. Consider the following function
> > > $$
> > > h(\eta) = \inf_{x,\varepsilon} [\varepsilon : \mathrm{dist}(x,x*) > \eta, \|\nabla f(x)\|\le\varepsilon, -\lambda_{\min}(\nabla^2 f(x)) \le \varepsilon],
> > > $$
> > > We say a point $x$ is $h(\eta)$-second-order optimal if it satisfies $\|\nabla f(x)\|\le h(\eta)$ and $\lambda_{\min}(\nabla^2 f(x))\geq -h(\eta)$.
> > > By definition, every $h(\eta)$-second-order optimal point is $\eta$-close to the ground truth.
> > > * The reviewer's proof shows that $h(\eta)>0$ for every $\eta>0$. However, the required accuracy $h(\eta)$ can be arbitrarily small for any useful value of the radius $\eta>0$ without violating this fact. This is the "flatness" of the landscape alluded to in our previous response. Such flatness arising from the small values of $h(\eta)$ may cause the local search algorithms, such as perturbed gradient descent, to stagnate around saddle points (see e.g. Jin et. al. [R2]).
> > > * Equivalently, let $\rho$ be an upper-bound on the inverse of $h$, meaning that $\eta\le\rho(h(\eta))$. Then, every $\varepsilon$-second-order optimal point is $\rho(\varepsilon)$-close to the ground truth. The reviewer's proof shows that $\rho(\varepsilon)$ exists and is finite for every $\varepsilon>0$. However, the radius $\rho(\varepsilon)$ can be arbitrarily large for any nontrivial value of $\varepsilon>0$.
> > > * We can contrast this with the case of exact rank $r=r^*$ with a suitable RIP constant. Here, Ge et al. [R3] proved the linear upper-bound $\rho(\varepsilon) \le 20 \varepsilon / \sqrt{\lambda_r(M^*)}$ in establishing the strict saddle property. Using this bound, they proved that every $\varepsilon$-second-order optimal point is $O(\varepsilon)$-close to the ground truth.
> > > *  In the over-parameterized case, the linear upper-bound becomes vacuous. While the reviewer's proof shows that $\rho(\varepsilon)$ exists and is finite, we additionally require a tractable relationship between  $\rho(\varepsilon)$ and $\varepsilon$ so that the strict saddle property is useful in practice. In particular, $\rho(\varepsilon)$ should not "blow up" as $\varepsilon$ increases from zero.
> > >
> > > [R1] Tong, Tian, Cong Ma, and Yuejie Chi. "Low-rank matrix recovery with scaled subgradient methods: Fast and robust convergence without the condition number." IEEE Transactions on Signal Processing 69 (2021): 2396-2409.
> > >
> > > [R2] Jin, Chi, et al. "How to escape saddle points efficiently." International Conference on Machine Learning. PMLR, 2017.
> > >
> > > [R3] Ge, Rong, Chi Jin, and Yi Zheng. "No spurious local minima in nonconvex low rank problems: A unified geometric analysis." International Conference on Machine Learning. PMLR, 2017.

---

> > > > ### Comment · Reviewer_KAr6 · 2021-08-17
> > > > **Clarifications**
> > > >
> > > > Dear authors,
> > > >
> > > > Thank you very much for your response.
> > > >
> > > > Regarding the first point, I am not sure I understand your reply. Are you suggesting that the result in [R1] is not correct? Or is it your code which is not correct? Otherwise, how do you explain the discrepancy between the theoretical result and your numerical experiments? And how do you plan to update Figure 2 to show that ScaledGD fails because of overparameterization if you can't make it work in the exactly parameterized case?
> > > >
> > > > Regarding the second point, I agree that local search algorithms are slower in the overparameterized case. My point is that it is not correct to say that they are slower because the strict saddle property is not satisfied: the strict saddle property *is* satisfied. However, as you say, the dependency between the constants in the definition of the strict saddle property is much worse than in the exactly parameterized setting, which makes algorithms slower.
> > > >
> > > > As a remark, $\rho$ does not "blow up": it is a non decreasing function, so it is bounded on any bounded interval (for instance, $\rho(\epsilon) \leq \rho(1)$ for any $\epsilon\in[0;1]$). In addition, $\rho(\epsilon)\to 0$ when $\epsilon \to 0$.

---

> > > > > ### Author Response · Authors · 2021-08-17
> > > > > **Re: Clarifications**
> > > > >
> > > > > We apologize for the awkward wording in our previous response, which led to more confusion.
> > > > >
> > > > > >In our own experiments, ScaledGD continues to fail even with the specific step-sizes given in Tong et al. [R1] for the non-smooth, exactly parameterized case
> > > > >
> > > > > In the above quote, we meant to say that, for nonsmooth loss functions, we took the Tong et al. step-size scheme, which was developed for $r=r^*$, applied it to $r>r^*$, and observed failure. We are of course not suggesting that [R1] is incorrect for $r=r^*$. Indeed, our experiments found that when $r=r^*$, the step-size scheme works as intended. Our point is only that in the case of $r>r^*$, then the step-size scheme also exhibits erratic behavior.
> > > > >
> > > > > We will include an update to Figure 2 in which the top row has the exactly parameterized $r=r^*$ cases, and the bottom row has the over-parameterized $r>r^*$ cases. For ScaledGD, we will update the figure to use the step-size scheme in [R1]. For PrecGD, we will maintain the use of a constant step-size. The top row will show smooth convergence for all three $\ell_p$ losses, with comparable performance between ScaledGD (with special step-sizes) and PrecGD (with constant step-size). The bottom row will show similar behavior in PrecGD, but erratic behavior in ScaledGD. The top-bottom comparison will be comparable to Figure 1.
> > > > >
> > > > > We fully agree with the reviewer regarding the second point.

---

> > > > > > ### Comment · Reviewer_KAr6 · 2021-08-17
> > > > > > **Thank you**
> > > > > >
> > > > > > Dear authors,
> > > > > >
> > > > > > Thank you very much for your clarifications. I am sorry I did not understand the first part of your explanation right away, but everything is now clear.
> > > > > >
> > > > > > Best regards,
> > > > > > KAr6

---

### Official Review · Reviewer_bQAG · 2021-07-16

**Rating:** 6
**Confidence:** 4

**Summary:**

This paper studies the problem of (noisy) nonconvex matrix factorization. The authors proposed an algorithm PrecGD that accommodates to over-parameterization and ill-conditioned ground truth.

**Limitations And Societal Impact:**

This is a theoretical work and I think there is no potential societal impact.

**Main Review:**

Strengths:
1. The paper proposes a novel algorithm that performs well in numerical experiments, accompanied with solid theoretical guarantees.
2. The paper provides detailed high-level ideas and intuitions of their theory, which is helpful for those interested in their proof technique.

Weakness / comments:
1. The writing of this paper is not clear. Taking Section 5-6 as an example, I think it would be better to have a Section only presenting main results (Corollary 5, Proposition 6, Theorem 8), and this section should be self-contained. The intuitions / high-level ideas and the proof outlines can be deferred to earlier / later sections. The current version makes it difficult for readers to read the main result like Corollary 5 because necessary definitions required to understand Corollary 5 is spread out across the lengthy Section 5.
2. The definition of $\sigma$ first appeared in Theorem 7, but is already used in Proposition 6 and earlier in the main text. A definition is necessary before using this quantity.
3. It seems that Theorem 7 allows the noise level $\sigma$ to be arbitrarily large. I am skeptical about this as prior literature on matrix sensing usually prohibits the noise level from being too large, which usually reads $\sigma\sqrt{n/m}<<1$ under the setting of the current paper.
4. How to obtain an approximation of the variance in Theorem 8? What is the order of $\mathcal{E}_{var}$ in (21) under this approximation method? How does this quantity compared with two other quantities in (21)?
5. If I understand the remarks under Theorem 8 correctly, when the sample size $m$ is large, the estimation error rate becomes sub-optimal. This is quite strange as large sample size, or large SNR, ususally makes the problem easier. Could you explain more details on why you can get an optimal rate when SNR is small, but fail to do so when SNR is large?
6. I think maybe it is possible to estimate the rank precisely under the setting of Proposition 6. In that case, we no longer need to over-parameterize the model. Is that correct or not?

Typos:
Theorem 4 is sometimes cited as "Lemma 4".

**Time Spent Reviewing:**

2

---

> ### Author Response · Authors · 2021-08-10
> **Response to reviewer bQAG**
>
> We thank the reviewer for the valuable feedback and thoughtful comments. Please see our response below.
> > The writing of this paper is not clear.
>
> * We agree that in the current version the presentation of the main results can be improved. We will put more emphasize on the main results and defer some of the high-level ideas.
>
> > The definition of $\sigma$ first appeared in Theorem 7, but is already used in Proposition 6 and earlier in the main text. A definition is necessary before using this quantity.
> * We thank the reviewer for spotting this. We will fix this problem in the updated version.
>
> > It seems that Theorem 7 allows the noise level  to be arbitrarily large. I am skeptical about this as prior literature on matrix sensing usually prohibits the noise level from being too large, which usually reads  under the setting of the current paper.
>
> * We would like to point out that in Theorem 7, the final error bound depends on $\sigma\sqrt{n/m}$ (see the definition of $\mathcal{E}_{stat}$). Therefore, a large value of $\sigma$ relative to $\sqrt{n/m}$ leads to a rather loose and even vacuous bound on the final error, as expected. We also note that the same error term appears in other relevant papers (see e.g. [14], [33], [54]), and is known to be minimax optimal up to logarithmic factors (see [6]).
>
> > How to obtain an approximation of the variance in Theorem 8? What is the order of $\mathcal{E}_{var}$ in (21) under this approximation method? How does this quantity compared with two other quantities in (21)?
>  * We thank the reviewer for asking this question. We would like to point out that our contribution is in demonstrating the existence of a GD-like algorithm that is fundamentally immune to the ill effects of over-parameterization/ill-conditioning. In the noisy setting, we prove the existence of a range of damping coefficients that guarantees a linear convergence to the minimax error bound (see Theorem 7). This requires not only a careful balance between the gradient dominance condition that $\|\nabla f(X)\|^2\geq \mu f(X)$ and the Lipschitz-like inequality in Lemma 2, but also a guarantee that these conditions still hold under noise. The existence of such an algorithm opens the door for more practical algorithms in the face of noise.
> * As a first step towards achieving this goal, we study the performance of our algorithm under the assumption that an estimation of the noise variance is available. Evidently, obtaining such estimation is not the main focus of this paper, and necessitates an independent study of its own. However, to address the reviewer's comment, we briefly explain one possible approach to estimate the noise variance. Recall that the available measurements are of the form $y = \mathcal{A}(M^*)+\varepsilon$, where $\mathcal{A}(\cdot)$ is a linear mapping, and $\varepsilon$ is the noise vector. Ignoring the low-rank nature of $M^*$, one can rewrite the problem as an instance of linear regression, i.e., recovering a vector $\mathbf{x}$ (resembling the vectorization of $M^*$), from a limited number of linear measurements of the form $y = \bar{\mathbf{A}}\mathbf{x} + \varepsilon$. Indeed, such simplification comes at the expense of losing the low-rank nature of the solution, but it can be used as a cheap method to estimate the noise variance via several well-known methods in linear regression, including scaled LASSO [B1], refitted cross-validation [B2], and moment-based estimator [B3], to name a few. We consider a rigorous study of this practical estimation method, including its sample complexity, as an enticing challenge for future research. We will add this discussion to the main body of the paper.
>
> > If I understand the remarks under Theorem 8 correctly, when the sample size  is large, the estimation error rate becomes sub-optimal. This is quite strange as large sample size, or large SNR, ususally makes the problem easier. Could you explain more details on why you can get an optimal rate when SNR is small, but fail to do so when SNR is large?
>
> * We thank the reviewer for this insightful comment. We believe that there is a misunderstanding about the role of SNR in our final error. An improved SNR should naturally lead to a better estimation error. This is indeed the case in our derived bound in Theorem 8, where the final error bound improves with SNR (i.e., the number of samples). However, the rate of improvement in the error is suboptimal with an estimated variance (while still improving with an increasing SNR). To explain the main reason behind this suboptimality, recall that after a sufficient number of iterations, the final error is governed by the maximum of three terms: the minimax error $\mathcal{E}\_{stat}$, the deviation of the empirical variance from its true value $\mathcal{E}\_{dev}$, and the variance estimation error $\mathcal{E}\_{var}$. Moreover, for simplicity, let us assume that the variance is known precisely, i.e., $\mathcal{E}\_{var} = 0$. Our remark after Theorem 8 is intended to show that the suboptimality of the final error occurs precisely when $\mathcal{E}\_{dev}$ dominates $\mathcal{E}\_{stat}$. This happens when the number of samples exceeds a certain threshold. In other words, we have $\mathcal{E}\_{dev}\geq \mathcal{E}\_{stat}$ only if $m\gtrsim \sigma^2 n^2 r^2 \log n$. The slower decay rate of $\mathcal{E}\_{dev}$ is due to the heavy tail phenomenon cause by the concentration of the noise variance, which dominates the overall error only in the high-sampling regime. We will better clarify this point in the revised paper.
>
> > I think maybe it is possible to estimate the rank precisely under the setting of Proposition 6. In that case, we no longer need to over-parameterize the model. Is that correct or not?
>
> * We note that it is usually not possible to estimate the correct rank via spectral initialization. Consider the following simple example, where the ground truth is $M^\star$ and the initial point is $X_0$, as in
>  $
> M^\star = [
> 1  ~~0;
>  0  ~~0	]
> $, $X_0 = [
> 1-\rho ~~ 0;
>  0 ~~ \rho
> ],$
> and $\mathcal{A}$ is the identity operator (which trivially satisfies RIP with $\delta=0$). This is an example that satisfies the spectral initialization condition
> $$
> \|\mathcal{A}(X_0X_0^T-M^\star)\|^2 \leq (1-\delta) \cdot \rho^2 \cdot \lambda_{r^*}^2(M^\star)
> .$$
> Nevertheless, the rank of $M^\star$ cannot be known from simply looking at $X_0$. Indeed, if $\rho=\frac 1 2,$ then the two eigenvalues of $X_0$ coincide, and we cannot determine what the true rank is. We note that counterexamples for the more general setting, with arbitrary $r$, $r^*$, and $n$, can be constructed in an identical way.
>
> [B1] Sun, Tingni, and Cun-Hui Zhang. "Scaled sparse linear regression." Biometrika 99.4 (2012): 879-898.
>
> [B2] Fan, Jianqing, Shaojun Guo, and Ning Hao. "Variance estimation using refitted cross‐validation in ultrahigh dimensional regression." Journal of the Royal Statistical Society: Series B (Statistical Methodology) 74.1 (2012): 37-65.
>
> [B3] Dicker, Lee H. "Variance estimation in high-dimensional linear models." Biometrika 101.2 (2014): 269-284.

---

> > ### Comment · Reviewer_bQAG · 2021-08-31
> > **Two remaining questions**
> >
> > Thanks the authors for detailed response. I still have two remaining questions after reading the response.
> >
> > 1. In Equation (21), $\sigma$ is quadratic in the definition of $\mathcal{E}{\mathsf{stat}}$ but linear in the definition of $\mathcal{E}{\mathsf{dev}}$, which is strange. I guess that $\sigma^2$ in the definition of $\mathcal{E}\mathsf{dev}$ should not be in the square root. This is because the "variance deviation" term in Equation (22) can be upper bounded by the order of $\sigma^2\sqrt{\frac{\log(n)}{m}}$ using Bernstein's inequality. If this term can be upper bounded by $(1-\delta)\mathcal{E}{\mathsf{dev}}$, as stated in line 299, then the definition of $\mathcal{E}{\mathsf{dev}}$ should be quadratic in $\sigma$. I also suggest the authors to check the definition of $\mathcal{E}\mathsf{var}$ in Equation (21) which is quartic in $\sigma$. I guess it should be either quadratic in $\sigma$, or divided by something like $\lambda_{r^\star}^2(M^\star)$.
> >
> > 2. I am confused with the response on the possibility of using spectral initialization to estimate the rank. Given the ground truth is $M^\star=[1,0;0,0]$ and $\mathcal{A}$ is identity, where does $X_0=[1-\rho,0;0,\rho]$ come from, and why not use Equation (18) to construct $X_0$? In my opinion, it is straightforward to use matrix perturbation theory (Davis-Kahan Theorem) to show that the top $r$ eigenvalues of $X_0$ are spiky and much larger than the remaining eigenvalues, under the RIP condition and the noise condition (in noisy setting) in the paper.
> >
> > Overall, currently I would like to keep the score unchanged.

---

> > > ### Author Response · Authors · 2021-09-01
> > > **Response to additional questions**
> > >
> > > We thank the reviewer again for these insightful questions. Please see our response below.
> > >
> > > **Response to comment 1:** We thank the reviewer for pointing out these typos. The correct definitions of $\mathcal{E}\_{dev}$ and $\mathcal{E}\_{var}$ are $\frac{\sigma^2}{1-\delta}\sqrt{\frac{\log m}{n}}$ and $|\sigma^2-\hat\sigma^2|$, respectively. These were derived and stated correctly in our proof in Section G of Appendix, but were copied incorrectly into the main text. We will fix these typos in the revised manuscript.
> > >
> > > **Response to comment 2:** We agree with the reviewer that the exact rank $r^*$ is easily
> > > recovered using spectral initialization if a large spectral gap exists between the smallest singular
> > > value of the ground truth $\lambda_{r^*}(M^*)$ and the noise
> > > variance $\sigma$. While gradient descent may converge sublinearly with over-parameterization $r>r^*$, once the exact rank $r^*$ is recovered, we can simply set $r=r^*$ and then expect rapid convergence. This is essentially the *well-conditioned*
> > > regime, in which the condition number $\lambda_{1}(M^*)/\lambda_{r^*}(M^*)$
> > > is assumed to be modest, and we do not claim any novelty here since gradient descent already works well.
> > >
> > > However, in the ill-conditioned case, it is not always possible to estimate the ''correct'' rank $r^*$ using spectral initialization. Moreoever, even if the correct rank were known, setting $r=r^*$ may not necessarily lead to rapid convergence, due to the effects of ill-conditioning.
> > >
> > > For example, let $\lambda_{k}(M^*)=1/k$ for $k=1,\dots,r^*$. For reasonably large $r^*$, the gap between $\lambda_{r^*}(M^*)$
> > > and $\sigma$ is much smaller than the magnitude $\lambda_{1}(M^*)$,
> > > so we do not expect Davis-Kahan to be predictive. The problem of estimating the ''exact rank'' becomes more and more difficult as the spectral gap becomes smaller.  Indeed, the problem
> > > is not even well-defined once the gap goes to zero. Moreover, getting the ''right'' rank does not help. Even if we assume that $r=r^*$ were chosen correctly, gradient
> > > descent would still converge very slowly due to ill-conditioning. In fact, the convergence rate can become arbitrarily slow as the condition number $\lambda_{1}(M^*)/\lambda_{r^*}(M^*)$ goes to infinity.
> > >
> > > An important feature of PrecGD is that it corrects for both ill-conditioning
> > > and over-parameterization at the same time, without viewing them as distinct concepts. In the above example, with a correctly chosen rank $r=r^*$, PrecGD  corrects for ill-conditioning, and converges quickly as if the
> > >  ground truth were perfectly conditioned $\lambda_{1}(M^*)=\lambda_{r}(M^*)$. But even if
> > > the rank $r\ge r^*$ were selected incorrectly, PrecGD corrects for over-parameterization, and will converge at the same rapid rate, as if the ground truth were perfectly conditioned up to this incorrect rank.

---

> > > > ### Comment · Reviewer_bQAG · 2021-09-03
> > > > **Thanks for the response**
> > > >
> > > > I thank the authors for responding to my additional question. I will update my score.

---

### Official Review · Reviewer_hWq7 · 2021-07-20

**Rating:** 6
**Confidence:** 4

**Summary:**

This paper proposed a preconditioned gradient descent (PreGD) method for low rank matrix factorization.
In practice, the true rank r* of the matrix is unknown.
To be conservative, people usually choose an over-estimated r to start with.
However, in such case, the convergence of local search algorithms falls from a linear rate (r=r*) to a sublinear rate (r>r*).
The PreGD method proposed in this paper restores the linear convergence rate for the r>r* case.
Numerically, it is also found that the PreGD method works well in restoring the linear convergence for some variants of the problem.

**Main Review:**

The major contributions of this paper are twofold:
(1) the so called PreGD method;
(2) It is shown that PreGD restores the linear convergence for an over-estimated rank.

General Comments:

(1) The PreGD method, in my opinion, is not surprising, since it is quite natural to impose a regularizer \eta to cure the ill-conditioning for the over-estimated case. In addition, in practice, after a few iterations of GD, the true rank r* can be revealed from the SVD of the current estimation of X, under the assumption that lambda_{r*}(M*) is larger than the noise. So, one may simply run GD for a few iterations with an overestimated rank at the beginning stage, then reveal the true rank r*, after that, continue to run GD with the true rank. So, the value of PreGD in practice is not significant.

(2) The major contribution of this paper is (2) -- the theoretical analysis. The major results (corollary 5, Theorems 7 and 8) are all established under the assumption that a good initial guess of X is obtained from spectral initialization such that (17) holds. Let us consider the case when lambda_{r*} is small. The smaller lambda_{r*}, the more samples are needed to ensure (17) (by proposition 6). However, when M* has a low numerical rank that is much smaller than the true rank, the condition on m in proposition 6 will be too restrictive. Note that the overestimated rank case is quite similar to the low numerical rank case, I believe the results in this paper can be generalized to deal with the low numerical case, the term lambda_{r*}(M*)  in the  lower bound for m in proposition 6 will be replaced by some other eigenvalue of M* larger than a threshold.

(3) Figure 2, why the error for PreGD stagnates? Is it because of the noise? This is unclear from the context.


Minor comments:

Some tyros need to be fixed, e.g.,

Line 227, equation (16), a full stop is missing.

Line 233, Lemma 4 —> Theorem 4 ?

Line 237, theorem 5 —> corollary 5 ?



**Time Spent Reviewing:**

4

---

> ### Author Response · Authors · 2021-08-10
> **Response to reviewer hWq7**
>
>  We thank the reviewer for the valuable feedback and excellent questions.  Please see the following response to your comments.
> > The PreGD method, in my opinion, is not surprising, since it is quite natural to impose a regularizer $\eta$ to cure the ill-conditioning for the over-estimated case.
>
> * We agree with the reviewer that adding a regularizer is not surprising. We would like to emphasize that our contribution is in demonstrating the possibility of striking a perfect balance between ScaledGD and GD with a good choice of the regularization parameter $\eta_k$. Indeed, it is unclear a priori whether gradient dominance could be maintained if a regularizer were added. At the same time, without sufficient regularization, the over-estimated case would remain too ill-conditioned to avoid sporadic behavior.
>
> * Surprisingly, we find that gradient dominance is maintained for a small $\eta_k \lesssim \|X_k X_k^T - M^{\star}\|_F$, while it take at least $\eta_k \gtrsim  \|X_k X_k^T - M^{\star}\|_F$ to avoid sporadic behavior. The two ranges of $\eta_k$ intersect, and there exists a choice of $\eta_k \approx \|X_k X_k^T - M^{\star}\|_F$ to allow both properties to be maintained. While the basic principle of regularization is very natural, proving the existence of an $\eta_k$ that lies at this intersection is difficult, and is the main contribution of our paper.
> The resulting GD-like algorithm is fundamentally immune to the issues of over-parameterization/ill-conditioning, and is guaranteed to converge linearly to the ground truth up to minimax optimal error.
>
> > In practice, after a few iterations of GD, the true rank $r*$ can be revealed from the SVD of the current estimation of X, under the assumption that $\lambda_{r*}(M^\star)$ is larger than the noise. The value of PreGD in practice is not significant.
> * This is actually not the case. Consider a simple example where the noise is zero and the eigenvalues of the ground truth are $\lambda_k = 1/k$ for $k=1,\dots, r^*$. Since GD converges sublinearly, after just a few iterations, we have no way of determining whether the smaller singular values of $X_k$ are the ones corresponding to the ground truth or the ones  that simply have not converged to 0 yet. As a result, we cannot know the rank of the ground truth by looking at $X_k$.
>
> * Even in the case that the ground truth is well-conditioned, it still takes a long time for gradient descent to reveal the true rank. The main reason, as shown in the recent results of Zhuo et al. [54], is that the error spreads over all ranks, and it is actually that error in the over-specified rank that causes GD to stagnate.
>  As a result, we do not see the true rank of the ground truth until gradient descent almost converges. Therefore, over-parameterization cannot  be easily avoided, and PrecGD is much faster than GD in this regime.
>
> > The results in this paper can be generalized to deal with the low numerical case, the term $\lambda_{r*}(M^\star)$ in the lower bound for m in proposition 6 will be replaced by some other eigenvalue of $M^\star$ larger than a threshold.
>
> * We thank the reviewer for pointing out the similarities between the low numerical rank case and the over-parameterized case. We agree that the sample complexity in proposition 6 might be too restrictive in the low numerical rank case and that our results can be made less conservative for this case. In fact, as the reviewer points out, our analysis for the noisy case can easily accommodate the low numerical rank case, if we treat the smallest singular values of the ground truth as noise. We will update our paper to reflect these suggestions.
>
> > Why does PrecGD stagnate in Figure 2?
> * In our numerical experiments for non-smooth functions, we used a constant step-size in order to compare the performance of GD and PrecGD. In general, for non-smooth functions (that do not satisfy gradient Lipschitzness by construction), a diminishing step-size is necessary for avoiding stagnation. This has been demonstrated rigorously for ScaledGD (see [A1]). We believe that similar conditions hold for PrecGD. However, understanding the performance of PrecGD in a non-smooth setting is not the main focus of this paper; it is left as future work.
>
> [A1] Tong, Tian, Cong Ma, and Yuejie Chi. "Low-rank matrix recovery with scaled subgradient methods: Fast and robust convergence without the condition number." IEEE Transactions on Signal Processing 69 (2021): 2396-2409.
>
> [54] Jiacheng Zhuo, Jeongyeol Kwon, Nhat Ho, and Constantine Caramanis. On the computational and statistical complexity of over-parameterized matrix sensing. arXiv preprint arX345 iv:2102.02756, 2021.

---

### Decision · Program_Chairs · 2021-09-27

**Decision:**

Accept (Poster)

**Comment:**

The paper proposes regularized scaled gradient descent for low-rank matrix factorization. Particularly, it considers the scenarios where the model parameter r is an overestimation of the true rank r*. This is quite a common scenario. Though the scaling (and its regularized variant) is well-known (see papers from 2013 onwards based on numerical analysis or Riemannian metric selection), a deeper convergence analysis was missing that the paper presents. Though experiments are adequate for showing the good performance, some comparisons on real data would have been nice. Nevertheless, the reviewers agree that the paper has useful insights for NeurIPS community.